# ONLINE RANKING WITH UNFAIR FEEDBACK AND HUMAN VERIFICATION:
# HIERARCHICAL ELIMINATION AND REGRET BOUNDS

## ABSTRACT

Online platforms rely heavily on user feedback for ranking systems, such as restaurant ratings and e-commerce listings. However, these systems face challenges from unfair feedback, including merchant-induced and malicious feedback. Thus, platforms have adopted human verification to increase the reliability of the rankings. It can certainly identify genuine feedback, but introduces high latency into real-time updates, leading to non-static queuing dynamics and creating challenges for online learning. We model this as a continuous-time online learning problem, establish the lower bound on regret, and propose two algorithms: Hierarchical Elimination (HE) and Deficit Hierarchical Elimination (DHE), dealing with the case of single and multiple verifiers, respectively. We further prove upper regret bounds for both algorithms and demonstrate their effectiveness through numerical experiments.

## 1 INTRODUCTION

The pervasive influence of online ranking systems has made them crucial components of modern digital platforms, serving as essential tools for content discovery and decision-making across various domains including e-commerce, content sharing, and service platforms (Golrezaei et al., 2023; Negahban et al., 2017). These systems typically rely heavily on user feedback to determine rankings, operating under the assumption that such feedback accurately reflects item quality. However, this assumption has been increasingly challenged by the prevalence of unfair feedback - reviews or ratings that deliberately misrepresent item quality due to various motivations including competitive manipulation, personal bias, or financial incentives.

Recent developments in major online platforms have introduced verification mechanisms to address this challenge. For instance, some platforms such as Meituan have implemented jury-like review panels that verify the authenticity and fairness of user feedback (see Appendix C). These panels examine suspicious reviews through various means including on-site verification, photographic evidence, and cross-referencing with transaction records. While such mechanisms show promise in maintaining ranking integrity, they introduce new theoretical challenges that existing frameworks are ill-equipped to address, such as how verification rate and policy impact the online ranking accuracy, or if it is possible to extract information from unverified feedback in overloaded systems.

The primary challenge lies in the inherent tension between verification thoroughness and system responsiveness. Verification mechanisms, while crucial for maintaining ranking accuracy, introduce delays in feedback processing. These delays create complex queuing dynamics that interact with the learning process in non-trivial ways. Moreover, the system must operate continuously, making real-time ranking decisions while simultaneously learning from both verified and unverified feedback. This creates a complex online learning problem where the learning process is intimately coupled with the underlying queuing dynamics.

Our main contribution is the development of a comprehensive framework for dynamic ranking systems with unfair feedback and verification mechanisms. We propose the Hierarchical Elimination (HE) algorithm that achieves logarithmic regret bounds by effectively utilizing both verified and unverified feedback, and extend it to the Deficit Hierarchical Elimination (DHE) scheduling policy for

systems with multiple heterogeneous verifiers. Through rigorous theoretical analysis, we establish bounds on system regret, demonstrating the effectiveness of our proposed algorithms.

The rest of the paper is organized as follows. Section 2 reviews related work across multiple domains. Section 3 presents our model and formally defines the optimization objective. Section 4 introduces our main algorithms and provides theoretical guarantees. Section 5 extends the analysis to multiple heterogeneous verifiers. Section 6 presents experimental results validating our theoretical findings, and Section 7 concludes with discussions of future directions.

## 2 RELATED WORK

The field of bandit algorithms has developed a rich theoretical foundation, built upon seminal algorithms such as UCB (Auer, 2002; Lai & Robbins, 1985), LinUCB (Abbasi-Yadkori et al., 2011)), and SE (Successive Elimination) (Even-Dar et al., 2006). This framework has been extended to accommodate complex user behavior through customer choice models, as exemplified by the bandit-MNL approach (Agrawal et al., 2018) and choice bandits (Agarwal et al., 2020). While these advances are significant, they primarily address subset selection problems, leaving the challenges of item ranking relatively unexplored.

The specific problem of online learning to rank has evolved along a parallel trajectory, with notable contributions from (Zoghi et al., 2017; Li et al., 2019; Lattimore et al., 2018), establishing fundamental frameworks, and subsequent works incorporating click models such as PBM (Lagrée et al., 2016) and cascade models (Kveton et al., 2015). However, these approaches predominantly optimize for click-through rates rather than comprehensive user utility metrics. A particularly relevant recent advancement (Zuo et al., 2023) addresses the critical issue of adversarial feedback attack, though their reliance on stylized attack models and stationarity assumptions potentially limits real-world applicability. The work (Golrezaei et al., 2022) focuses on traditional learning-to-rank aiming for maximizing click-through rates, while our work is concerned with maximizing the consumer's true experience–a setting more aligned with multi-armed bandits. In addition, their robust algorithm deals with fake clicks under the assumption that the operator cannot verify the authenticity of the feedback. In contrast, our work is motivated by real-world scenarios (e.g., Meituan) and investigates how a verification system can be designed to integrate verification strategies with online learning.

The introduction of verification mechanisms, while crucial for feedback validation, introduces an inherent delay component to the learning process. This intersects with delayed feedback literature that have been extensively studied (Joulani et al., 2013; Dudik et al., 2011; Gael et al., 2020; György & Joulani, 2020; Lancewicki et al., 2021). The comprehensive study (Lancewicki et al., 2021) yields important insights of the superiority of successive elimination over UCB in delayed feedback scenarios. The concept of "soft delays" in (Esposito et al., 2023), where intermediate observations during delay periods containing valuable information provides a paradigm that naturally extends to our setting where even potentially unfair feedback carries information. However, the delay between intermediate observation and final feedback is not predetermined; specifically, it is governed by the verification policy, which necessitates consideration of queuing dynamics.

However, the studies of online learning in queuing systems primarily focus on system stability rather than user utility. Moreover, static system dynamics is commonly assumed. For example, the work (Huang et al., 2023) examines the impact of learning on system steady-state behavior, and the work (Krishnasamy et al., 2016) provides queue-length regret bounds, while this work (Krishnasamy et al., 2019) addresses the challenges of service rate learning. Whereas ranking systems present unique challenges due to position-dependent arrival rates, which fundamentally alters the system dynamics and demands novel theoretical frameworks and solutions.

## 3 MODEL AND OBJECTIVE

This section presents our model framework in two parts. First, we define the three key components of our dynamic ranking system: the ranking system, customer behavior patterns, and the verification process. Subsequently, we formalize the optimization objective and characterize the system dynamics that govern the learning process.

## 3.1 PROBLEM FORMULATION

The ranking system with unfair feedback and human verification comprises three key components:

**Ranking System.** We consider a system with a total of $K$ items, denoted by $\mathcal{I} = \{I_1, I_2, \ldots, I_K\}$, which are to be ranked from $1$ to $K$. Each item has an inherent quality parameter $\beta = \{\beta_1, \beta_2, \ldots, \beta_K\}$. At any time $t$, the operator can dynamically change the ranking of any item. We denote $\beta(t)$ as the vector of quality parameters ordered according to the current ranking at time $t$. It is a vector of dimension $K$ and each permutation of elements in this vector represents a possible ranking. Without loss of generality, we assume that the items are initially ordered such that $\beta_1 > \beta_2 > \cdots > \beta_K$. For simplicity, we display all items, though our algorithm can easily extend with the same order of regret when only a subset is displayed.

**Customer Behavior.** Customer arrivals follow a Poisson process with a rate normalized to 1. Usually, customers do not have prior knowledge about classes of highly similar products (e.g., standardized products like coffee rankings, electronic items like USB cables, or homogeneous services such as weather apps or flashlight apps from an app store). Hence, they rely on the platform's intelligent ranking system to make their choices. We assume that upon arrival, a customer selects an item from the ranked list purely based on its current position. Specifically, the probability that a customer selects the item ranked at the $i$-th position is $\alpha_i$, and we assume $\alpha_1 > \alpha_2 > \cdots > \alpha_K$.

After selecting item $I_k$, the customer provides immediate binary feedback on the selected item. Specifically, there is a probability $\beta_k$ of receiving good feedback and $1 - \beta_k$ of receiving bad feedback. However, with probability $\phi_k$, the feedback from the customer is manipulated, and we refer to such feedback as unfair feedback. We further assume that the distribution of an unfair feedback is Bernoulli with unknown mean $q_k(t)$ for item $k$, indicating that the manipulation behavior is non-stationary and lacks analytical properties. Note that the dependency on $t$ creates flexibility for the "attack" behavior and especially useful when constructing lower bounds.

**Human Verification.** Since it is indistinguishable between unfair and fair feedback without verification, human verification is introduced to verify if the feedback is fair and its true value. Specifically, all feedback from item $I_k$ is sent to its corresponding queue FCFS (first-come first-serve) $Q_k$ awaiting verification. While there is one verifier that can verify feedback from any queue with identical verification rate $\mu$. In other words, the verification time is exponentially distributed with mean $\frac{1}{\mu}$. After each verification, the operator of the system will know the true value of that feedback. It is worth mentioning that naively verify all feedback according to its arrival time will lead to high inefficiency, and thus, the scheduling policy should be carefully designed.

## 3.2 POLICIES AND OBJECTIVE

In the dynamic ranking system, we will consider an online learning problem which learns the true parameters $\beta$ to minimize the total decision error made through a finite continuous time horizon $T$.

**Ranking Policy** We define ranking policy $\pi_r$ to be a function that map histories to $[0, 1]^K$. Equivalently, we use $\beta^{\pi_r}(t)$ to denote the quality parameters after permutation based on the ranked list. For example, when $K = 2$ and $\beta = \{0.5, 0.4\}$, and item $I_1$ is ranked on the second place while item $I_2$ is ranked at the first place at time $t$, we have $\beta^{\pi_r}(t) = [0.4, 0.5]$, which is a vector of dimension $K$.

**Scheduling Policy** We define ranking policy $\pi_s$ that maps the current state of the system to the index set $[K]$. The policy decides which feedback to be verified at time t, denoted by $S^{\pi_s}(t)$, while within each type, we follows first come first served to avoid selection bias.

Given any pair of the policies $\pi = (\pi_r, \pi_s)$, we aimed to minimize the expected regret. By the assumption of decreasing $\alpha_i$, the optimal decision is always ranked the items according to their $\beta_k$s in descending order. Therefore, we define the regret by time $T$ as:

$$Reg(T) := \mathbb{E}^\pi \left[ \int_0^T (\beta - \beta^{\pi_r}(t))^T \alpha \, dt \right], \tag{1}$$

where the expectation is taken with respect to the dynamic of the customer arrival and choice, which is dependent on the policy. The continuous form of regret has barely no difference compared to discrete ones in expectation since the arrival rate is normalized to be one.

**System Dynamics** Before presenting the algorithm design, we need to first characterize the system dynamic for any policy $\pi$ and understand the complex interdependency between system state and the arrival process.

We introduce the following notations: For item $I_k$, let $Q_k^\pi(t)$ denote the number of feedback waiting to be verified, $A_k^\pi(t)$ be the cumulative arrivals with $\lambda_k^\pi(t)$ be the corresponding arrival rate, $S_k^\pi(t)$ be the number of feedback under verification, and $D_k^\pi(t)$ be the number of cumulative departures. The system follows:

$$Q_k^\pi(t) = A_k^\pi(t) - D_k^\pi(t) - S_k^\pi(t), \qquad (2)$$
$$\lambda_k^\pi(t) = f(\text{rank}(I_k)), \qquad (3)$$

where $f(\cdot)$ is the function that map the current rank to the corresponding arrival rate, which essentially depends on the system state. Such level of complexity implies that it is impossible to solve the queuing system analytically. For convenience, we use the tuple

$$(\mathcal{A}(t), m(t), n(t), m^p(t), n^p(t), LCB(t), UCB(t))$$

to denote the system state, representing the order sets, the numbers of verified feedback, the numbers of total feedback, the numbers of verified positive feedback, the numbers of total positive feedback, and the confident intervals, of all items. We will provide more detailed explanations of them in the following sections.

# 4 Algorithms and Regret Bounds

We present our algorithmic solutions and theoretical analysis in four parts. First, we introduce the Hierarchical Elimination (HE) algorithm for ranking and scheduling. Second, we establish logarithmic regret bounds for this algorithm. Third, we demonstrate how unverified feedback can be effectively utilized when bounded unfairness is known. Finally, we derive fundamental lower bounds on achievable regret.

## 4.1 HE Algorithm

We describe our algorithm in two components: the HE ranking policy that maintains and updates hierarchical sets of items, and the HE scheduling policy that prioritizes items for verification. In our algorithm, we need statistical estimations on the quality parameters. Specifically, we denote $\hat{\beta}_k(t) = m_k^p(t)/m_k(t)$ to be the empirical mean (fraction of positive feedback in verified feedback) of the quality parameters of item $I_k$ at time $t$. We further construct a confidence interval centered at its empirical mean using a radius of $\sqrt{\frac{\gamma \log(T)}{m_k(t)}}$, and the interval is denoted by $[LCB_k(t), UCB_k(t)]$ of item $I_k$ at time $t$. The full algorithmic version is in Appendix **??**.

**HE Ranking Policy** It starts with $K$ order sets, $\mathcal{A}^1$ to $\mathcal{A}^K$. Initially, we have $\mathcal{A}^1 = \{I_1, \ldots, I_K\}$ and $\mathcal{A}^q = \emptyset$ for $q > 1$. The algorithm is triggered only by the change of the system state such as arrivals or departures, and the time $t^+$ denotes the updated time. When triggered, once there exist $UCB_i < LCB_j$ for some $i, j \in \mathcal{A}^q$, we will send item $I_i$ to $\mathcal{A}^{q+1}$, where such event is called an elimination. We will use the set $\mathcal{B}$ to denote the union of non-singleton order sets, while $\mathcal{B}^c$ is those items in singleton sets. We will always rank $\mathcal{B}$ before $\mathcal{B}^c$. Within $\mathcal{B}$, we rank in ascending order of the total arrivals for each item, while within $\mathcal{B}^c$, we rank according to their corresponding order set index in ascending order.

For example, when $\mathcal{A}^1 = \{I_1\}, \mathcal{A}^2 = \{I_2, I_3\}, \mathcal{A}^3 = \{I_4\}, \mathcal{A}^4 = \emptyset$ and $n(t) = [10, 9, 8, 7]$, the ranking is $\{I_3, I_2, I_1, I_4\}$. Since both $I_1$ and $I_4$ are in singleton sets, and $I_1$ has smaller index (the index of $\mathcal{A}^1$ is 1), they ranked the third and the fourth. Also, since $I_3$ has smaller total feedback quantity, it ranked the top.

**HE Scheduling Policy** Priority is given to the item in $\mathcal{B}$ and contains the smallest number of verified samples, breaking tie arbitrarily.

## 4.2 Regret Analysis

Recall the definition of the regret, we need to bound the expected time where the rank is finalized, i.e., all sets all singleton, while prove that the probability that the final rank is incorrect is negligible.

**Theorem 1.** *Under the HE algorithm, the regret of the system*

$$Reg(T) \leq O\left(\sum_{k=1}^{K} \frac{\log(T)\,\Delta}{\min\{\Delta_{k-1,k}, \Delta_{k,k+1}\}^2 \mu} + \frac{\Delta}{\alpha_1}\right), \tag{4}$$

*where $\Delta_{k-1,k} := \beta_{k-1} - \beta_k$ is the gap between two consecutive items, and $\Delta = \sup_{\beta'}(\beta - \beta')^T \alpha$ is the largest possible regret rate. For handling the edge case, we define $\Delta_{K,K+1} = \Delta_{0,1} = 1$.*

The regret upper bound is composed of two parts: The first part arises due to the quality gap between items, which in a ranking system is quantified as the minimum gap between item $k$ and its adjacent item. The second part is due to the delay introduced by the queueing system, which is inversely proportional to the verification efficiency ($\mu$) and includes an initial queue delay of $\frac{1}{\alpha_1}$.

The proof of the theorem can be decomposed into the following steps: First, we claim that with high probability, the mean estimator for each item will lies on its confidence interval. Second, condition on this event, we bound the expected numbers of total samples for each arm before his rank is finalized. Next, due to the interdependency of the arrival rate and system state, it is intractable to find solve the departure processes for our system. Thus, we construct an less efficient system and show that the expected time before finalizing the rank is bounded by a logarithmic function with respect to $T$ for this system. Lastly, we show the expected time for finalizing the rank of the system operated using HE algorithm is less than the less efficient system.

### 4.3 Utilizing the Unverified Feedback

The previous algorithms utilize only verified feedback for ranking and scheduling decisions. This conservative approach stems from a fundamental statistical limitation: while we can construct mean estimators using both verified and unverified feedback, the confidence bounds for these estimators still depend critically on $m_k(t)$, the number of verified samples. This dependency arises because the uncertainty in the unfair feedback probability $\phi_k$ cannot be reduced without verification. Despite incorporating additional data points, current concentration inequalities do not yield faster convergence rates for confidence intervals constructed using unverified feedback.

However, when we have the information of an uniform upper bound on the unfair probability $\phi_k$, denoted by $\bar{\phi}$, we are able to construct three confident intervals for each item. For each item $I_k$, we define two additional quantities:

$$L\tilde{C}B_k(t) = \hat{\tilde{\beta}}_k(t) - \sqrt{\frac{\gamma \log(T)}{n_k(t)}}, \quad U\bar{C}B_k(t) = \hat{\bar{\beta}}_k(t) + \sqrt{\frac{\gamma \log(T)}{n_k(t)}}.$$

where the mean estimators $\hat{\bar{\beta}}_k(t)$ and $\hat{\tilde{\beta}}_k(t)$ are defined as:

$$\hat{\bar{\beta}}_k(t) = \frac{n_k^p(t) + n_k(t)\bar{\phi}}{n_k(t)}, \quad \hat{\tilde{\beta}}_k(t) = \frac{n_k^p(t) - n_k(t)\bar{\phi}}{n_k(t)}.$$

The $n_k^p(t)$ is the total number of positive feedback among all feedback no matter fair or unfair. We refer $\hat{\bar{\beta}}_k$ to be the super-optimistic estimation on item $I_k$, while $\hat{\tilde{\beta}}_k$ be the super-pessimistic estimation. As their name indicates, the super-optimistic estimation is an upper bound for the UCB constructed if we assume all feedback are verified, while super-pessimistic estimation serves as the lower bound.

Given the above quantities, we modify our elimination rule by adopting a bi-criteria rule where item $I_i$ is eliminated by $I_j$ when $UCB_i < LCB_j$ or $U\bar{C}B_i < L\tilde{C}B_j$. By such changes, the expected elimination time will be reduced for those items with their mean much smaller than the others. Specifically, we define the identifiable set

$$\Psi := \{I_k : (2\bar{\phi} + \phi_k + (1 - \phi_k)\beta_k < (1 - \phi_{k-1})\beta_{k-1})$$
$$\cup (2\bar{\phi} + \phi_{k+1} + (1 - \phi_{k+1})\beta_{k+1} < (1 - \phi_k)\beta_k)\}, \tag{5}$$

where henceforth, we define $\phi_0 = \phi_{k+1} = -\infty$ to handle edge cases. And for convenience, we further define $\delta_k = \min\{\delta_{k-1,k}, \delta_{k,k+1}\}$, and $\Delta_k = \min\{\Delta_{k-1,k}, \Delta_{k,k+1}\}$.

**Theorem 2.** *Under the existence of a known upper bound for unfair probability $\bar{\phi}$, such that $\phi_k < \bar{\phi}$ for all $k$. Under the HE algorithm with bi-criteria, the regret of the system*

$$
Reg(T) \leq O\Bigg( \sum_{k \in \Psi} \Delta \min \Big\{ \frac{\log(T)}{\Delta_k^2 \mu}, \frac{\log(T)}{\delta_k^2 \alpha_K} \Big\} \tag{6}
$$
$$
+ \sum_{k \notin \Psi} \frac{\Delta \log(T)}{\Delta_k^2 \mu} + \frac{\Delta}{\alpha_1} \Bigg),
$$

*where $\delta_{k-1,k} := (1 - \phi_{k-1})\beta_{k-1} - (2\bar{\phi} + \phi_k + (1 - \phi_k)\beta_k)$.*

Theorem 2 states that for items with larger gaps, the expected time for finalizing their rank is shorter. The proof is similar to the previous theorem, while in the last step, instead of constructing an single inefficient system, we decompose the original system into two less efficient systems and show that if both systems operate simultaneously, the expected time for finalizing the rank can upper bounded, and therefore, the expected time for the original system is also upper bounded.

### 4.4 LOWER BOUND

In this subsection, we will establish the lower bound by Theorem 3. The main challenges for deriving the lower bound are the followings. First, the complex interdependency between policies and stochastic queuing dynamics prevent the direct analysis. Second, it is challenging to quantify the information carried by unverified data.

**Theorem 3.** *Under any consistent algorithm satisfying Definition 1, the asymptotic regret of the system is lower bounded by*

$$
\liminf_{T \to \infty} \frac{Reg(T)}{\log(T)} \geq \Omega \left( \Delta_{min} \sum_{\xi=1}^{4} \sum_{k=1}^{K} \mathbf{1}\{I_k \in \Gamma_\xi\} C_k^\xi(\mu) \right), \tag{7}
$$

*where*

$$
\Gamma_1 = \{I_j : \phi_j \geq \tfrac{\Delta_{j,j+1}}{\Delta_{j,j+1}+1}, \ \phi_j \geq \tfrac{\Delta_{j-1,j}}{\Delta_{j-1,j}+1}\}, \tag{8}
$$

$$
\Gamma_2 = \{I_j : \phi_j < \tfrac{\Delta_{j,j+1}}{\Delta_{j,j+1}+1}, \ \phi_j < \tfrac{\Delta_{j-1,j}}{\Delta_{j-1,j}+1}\}, \tag{9}
$$

$$
\Gamma_3 = \{I_j : \phi_j \geq \tfrac{\Delta_{j,j+1}}{\Delta_{j,j+1}+1}, \ \phi_j < \tfrac{\Delta_{j-1,j}}{\Delta_{j-1,j}+1}\}, \tag{10}
$$

$$
\Gamma_4 = \{I_j : \phi_j < \tfrac{\Delta_{j,j+1}}{\Delta_{j,j+1}+1}, \ \phi_j \geq \tfrac{\Delta_{j-1,j}}{\Delta_{j-1,j}+1}\}. \tag{11}
$$

*and analytical form of $C_k^\xi(\mu)$ is presented in the appendix.*

The contributions of each item in the regret lower bound are grouped based on their likelihood of receiving unfair feedback and the quality gaps between them and their adjacent items. Each group has different information absorption capacity from unverified feedback. Specifically, for items that with small unfair probability and larger gaps between its adjacent items, the information of unverified feedback is potentially larger, vice versa. However, there is a minor gap between the lower bound and the upper bound in system parameters such as $\alpha_i$ due to the way we construct the coupling systems.

## 5 MULTIPLE VERIFIERS WITH HETEROGENEOUS RATES

In this section, we consider a more general setting where we have $N$ verifiers, where each verifier is denoted by $V_i$, and for verifier $V_i$, the verification rate for verifying item $I_j$ is $\mu_{ij}$. Given that heterogeneousness of verifiers, if we naively adopt the previous algorithm, the regret will be related to the minimum verification rate among all pairs, leading to inefficiency.

Furthermore, the assumption of preemption will be relaxed in this section. The reason behind it is that when there is only a single verifier, preemption or not will not affect the time for finalizing the rank. However, in multi-verifiers case, if we have $\mu_{ij}$ extremely small for some $i$ and $j$, then

verifying item $I_j$ using verifier $V_i$ will almost leads to a permanent deduction of verifier number by 1, for which it requires us to smartly idle the server when necessary.

**Example 5.1.** Consider the following instance, where $K = 4, N = 2$ and the system state are given by $\mathcal{A}^1 = \{I_1, I_2\}, \mathcal{A}^2 = \{I_3, I_4\}, \mathcal{A}^3 = \mathcal{A}^4 = \emptyset$ and $m(t) = [10, 8, 12, 16], n(t) = [12, 9, 17, 20]$. The verification rate $\mu_1 = [0.01, 1, 1, 1], \mu_2 = [1, 1, 0.01, 0.01]$. If both verifiers are idle now, verifier $V_2$ should verify the feedback of item $I_2$ given that $m_2(t)$ is the smallest, and $V_2$ is suitable for verifying $I_2$. However, for verifier $V_1$, there are several possible actions sounds reasonable. First, it can verify the feedback of item $I_1$ since there are no more feedback of item $I_2$ waiting in the queue, and $m_1(t)$ is the second smallest one. However, the verifier $V_1$ is not suitable for $I_1$, it may be a better decision to verify item $I_3$ or keep it idle to wait the next arrival of $I_2$. Furthermore, we not only need to decide the scheduling, but decide their ranking which is directly dependent on the arrival rates.

### 5.1 DEFICITS-BASED SCHEDULING POLICY

We aim to develop a scheduling policy that best aligns with the idea of elimination, and we define asymptotic optimality by maximizing the asymptotic minimum departure rate for feedback in the set $\mathcal{B}$. To formalize the scheduling policy, we introduce the decision variable $x_{ij}(t)$ representing whether verifier $i$ verifies feedback from item $I_j$.

**Assumption 1.** *We assume the system is overloaded such that $\sum_{i=1}^{N} \mu_{ij} < \frac{\alpha_1}{K}$ for any $j$.*

The assumption 1 states that the system is overloaded, where the total verification rate for any type of feedback is smaller than the top-item arrival rate. An immediate result from this assumption under our ranking policy is that there are always feedback waiting to be verified for any item, and therefore, we define the asymptotic max-min departure rate by the following relaxed linear programming:

$$\max_{x_{ij}} \min_j \sum_{i=1}^{N} x_{ij}\mu_{ij} \tag{12}$$

$$\text{s.t.} \quad \sum_{j \in \mathcal{B}} x_{ij} \leq 1, \quad \forall i, \tag{13}$$

$$x_{ij} \geq 0, \quad \forall i, j. \tag{14}$$

In the above LP, we allow partial allocation for each feedback, and by the overloaded assumption, it serves as an upper bound for asymptotic max-min departure rate. The solution of the LP is denoted by $x_{ij}^*(\mathcal{B})$ and the optimal value by $z^*(\mathcal{B})$, given the union of non-singleton sets, $\mathcal{B}$.

**DHE Scheduling Policy** Inspired by Deficit Round-Robin (Shreedhar & Varghese, 1996) algorithm in fair queuing systems, we proposed an scheduling policy based on deficits of the verification time, where we first solve the LP and get $x_{ij}^*(\mathcal{B})$ given the set $\mathcal{B}$, and we calculate the deficit $\theta_{ij}$ for each $(i, j)$ pair by the following definition

$$\theta_{ij}(t) = x_{ij}^*(\mathcal{B}) t_j - S_{ij}(t), \tag{15}$$

where $S_{ij}(t)$ is the total time that item $I_j$ has been verified by verifier $V_i$, and $t_j$ is the total verification time for verifier $V_j$. Once the server $V_i$ is empty, it will serve the item with the largest $\theta_{ij}(t)$, and has $x_{ij}^* > 0$. If there are any modifications on set $\mathcal{B}$ (elimination happens), we resolve the LP, and reset all deficits to 0. It is noticeable that the deficits accumulate only when the verifier is busy.

**Lemma 1.** *Under HE Ranking policy and DHE Scheduling policy and for a given set $\mathcal{B}$, the average deficit for any pair $(i, j)$ will converge to 0 if no elimination occurs.*

$$\limsup_{t \to \infty} \frac{\theta_{ij}(t)}{t} = 0 \tag{16}$$

*and for any finite $t$,*

$$\mathbb{E}\left[\frac{\theta_{ij}(t)}{t}\right] \leq \frac{\rho + \ln(\mu_{max}(\mathcal{B})t) + \text{Ei}(-\mu_{max}(\mathcal{B})t)}{\mu_{min}(\mathcal{B})t} \tag{17}$$

$$:= c(\mathcal{B}, t),$$

*where $\mu_{min}(\mathcal{B}), \mu_{max}(\mathcal{B})$ are the smallest (largest) verification rate for all pairs $(i, j)$ with $j \in \mathcal{B}$, $\rho$ is the Euler–Mascheroni constant, and $\text{Ei}(-x) = -\int_x^{\infty} \frac{e^{-t}}{t} dt$.*

The intuition behind this lemma is that for each verifier, the total deficits keep unchanged when the verifier is working, while each time a verification is completed, the maximum deficit for this verifier will generally decrease due to the control policy. Therefore, we are able to relate the deficits to the verification times, which are exponentially distributed.

## 5.2 REGRET UPPER BOUND

In the above lemma, we demonstrate the fair departure rate for any finite time without elimination. However, it is trivial to extend it to the case with elimination. The reason behind it is that after elimination, the minimum verification rate in the set $\mathcal{B}$ will no decrease, and we can still use exponential random variables to find its stochastic upper bound. Therefore, as an immediate result, we can derive the below upper bound of the regret for a finite time $T$ under our algorithm.

**Theorem 4.** *Under the existence of a known upper bound for unfair probability $\bar{\phi}$, such that $\phi_k < \bar{\phi}$ for all $k$. Under the HE algorithm with bi-criteria and DHE scheduling policy, the regret of the system is upper bounded by*

$$
\begin{aligned}
Reg(T) \leq O\Bigg( &\sum_{k \in \Psi} \Delta \min \Big\{ \frac{\log(T)}{\Delta_k^2 \big( z^*(\mathcal{I}) - c(\mathcal{I}, T) \big)}, \frac{\log(T)}{\delta_k^2 \alpha_K} \Big\} \\
&+ \sum_{k \notin \Psi} \frac{\Delta \log(T)}{\Delta_k^2 \big( z^*(\mathcal{I}) - c(\mathcal{I}, T) \big)} \Bigg).
\end{aligned}
\tag{18}
$$

Theorem 4 implies that the regret depends on the optimal fair queue departure speed in our system, which matches the intuition that when the more system can operate efficiently, the less the regret would be. It is also noticeable that $c(\mathcal{I}, T)$ is of order $\frac{\log(T)}{T}$, which is negligible for some large $T$.

## 6 NUMERICAL EXPERIMENTS

In this section, we perform experiments (on a single Nvidia i7-10700 CPU) of single verifier and multi verifiers to demonstrate the effectiveness of our algorithm in a simulated environment, and we also include some additional experiments such as the verification rate for each items in multi-verifier systems and the convergence of the deficits in Appendix B.

### 6.1 SINGLE VERIFIER

We begin with experiments in a single-server environment to illustrate the benefits of our bi-criteria elimination approach. Specifically, we perform two experiments: one utilizing the standard elimination criteria and another employing the bi-criteria method.

We consider a system with three items ($K = 3$) characterized by several key parameters. The quality parameters are set as $\beta = [0.9, 0.5, 0.1]$, with corresponding selection probabilities $\alpha = [0.7, 0.2, 0.1]$. We set uniform unfair feedback probabilities $\phi = [0.1, 0.1, 0.1]$ and positive feedback rates given unfair feedback as $q(t) = [0.7, 0.7, 0.7]$. The verification rate is fixed at $\mu = 0.4$. The system is simulated over a time horizon of $T = 2000$.

**Standard Elimination Criteria.** Figure 1(a) depicts the regret over time when using elimination based solely on confidence bounds constructed from verified samples. As expected from elimination algorithms, the regret grows linearly within each elimination phase. Initially, all three items are treated symmetrically, leading to an equal distribution of rankings and a corresponding average regret slope. Upon eliminating item $I_3$, the system proceeds with the remaining two items, resulting in a reduced slope corresponding to the regret rates of the rankings $[I_1, I_2, I_3]$ and $[I_2, I_1, I_3]$. Finally, after eliminating item $I_2$, only $I_1$ remains in a singleton set and is ranked last, causing an increase in the regret slope. The increase of the slope is unavoidable due to the queuing dynamics, if we rank item $I_1$ the top in the third phase, the effective arrival rate for identifying item $I_2$ and $I_3$ will be $1 - \alpha_1$, making them indistinguishable and leads to a long lasting linear regret accumulation.

**Bi-Criteria Elimination.** In contrast, Figure 1(b) demonstrates the advantage of the bi-criteria elimination approach. By leveraging both verified and unverified feedback and adopting a more

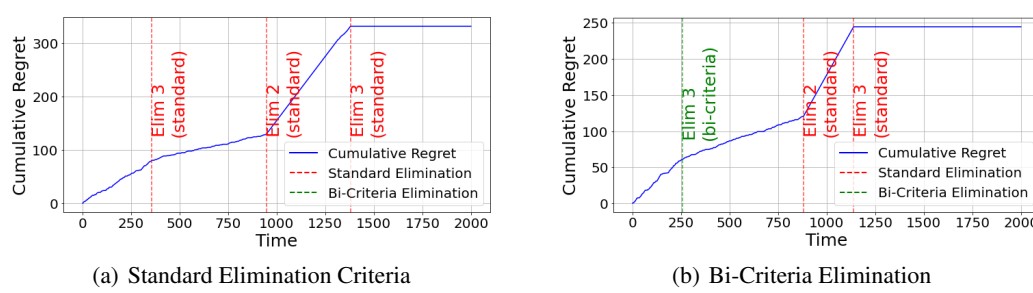

(a) Standard Elimination Criteria

(b) Bi-Criteria Elimination

Figure 1: Regret plots for single-server experiments.

conservative estimation strategy, the system achieves faster elimination of suboptimal items. This results in a lower cumulative regret, particularly noticeable when the verification rate is low and quality gaps are substantial.

## 6.2 MULTIPLE VERIFIERS WITH HETEROGENEOUS RATES

We consider a system with three items ($K = 3$) and two verifiers ($N = 2$) with the following configurations. The quality parameters are set as $\beta = [0.9, 0.5, 0.1]$, with selection probabilities $\alpha = [0.5, 0.3, 0.2]$. We maintain uniform unfair feedback probabilities $\phi = [0.1, 0.1, 0.1]$ and positive feedback rates given unfair feedback as $q(t) = [0.7, 0.7, 0.7]$. The verification rates vary by verifier, with verifier $V_1$ having rates $\mu_1 = [0.4, 0.15, 0.1]$ and verifier $V_2$ having rates $\mu_2 = [0.1, 0.15, 0.4]$. The simulation runs for $T = 1000$.

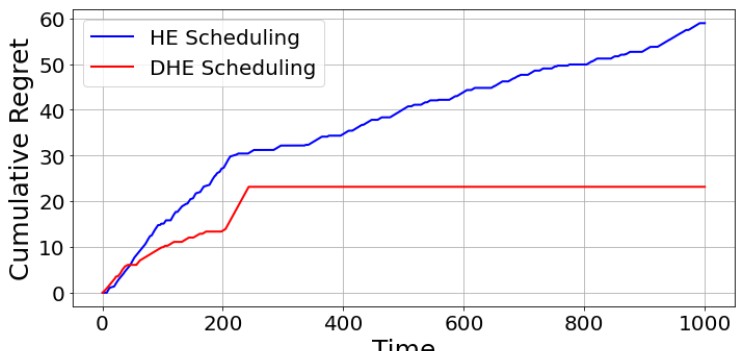

Figure 2: Regret comparison in multi-server experiments: HE scheduling vs. DHE scheduling.

**Regret Comparison.** Figure 2 compares the regret between the Hierarchical Elimination (HE) scheduling policy and our proposed Deficit Hierarchical Elimination (DHE) scheduling policy. The HE scheduling policy, which naively prioritizes items with the fewest verified feedback, exhibits inefficiency in this multi-verifier context. In contrast, the DHE scheduling policy effectively leverages the heterogeneous verification rates, resulting in lower cumulative regret.

## 7 CONCLUSION

We addressed ranking integrity challenges in online platforms affected by manipulated feedback by developing the Hierarchical Elimination (HE) algorithm for single-verifier systems and the Deficit Hierarchical Elimination (DHE) policy for multi-verifier environments. These algorithms effectively balance verified and unverified feedback, achieving logarithmic regret bounds. Future research directions conquering our limitations by developing algorithms with improved verification rate dependency, achieving item-specific regret rates, designing policies with minimal linear regret for better finite-time performance, and extending to contexts with unknown verification rates or contextual settings. Also, the study of G/G/c queue can also increase our applicability.

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

# A    PROOFS

## A.1    PROOF OF THEOREM 1

**Lemma 2.** *The conditional probability that the final rank is correct given the number of total verifications, $M_T$, is at least $1 - M_T T^{-2\gamma}$*

*Proof.* Define event $E$ to be the event that all true mean values are in the confidence interval for any time $t$.

$$E := \bigcap_{j=1}^{K} \{|\hat{\beta}_j(t) - \beta_j| \leq \sqrt{\frac{\gamma \log(T)}{m_j(t)}}\}, \text{ for all } t \tag{19}$$

By Hoeffding's inequality, we have

$$\mathbb{P}\left(|\hat{\beta}_j(t) - \beta_j| \leq \sqrt{\frac{\gamma \log(T)}{m_j(t)}}\right) \leq 2\exp(-2m_j(t)\frac{\gamma \log(T)}{m_j(t)}) \tag{20}$$

$$\leq \frac{1}{T^{2\gamma}} \tag{21}$$

By union bound,

$$\mathbb{P}(E^c) \leq M_T T^{-2\gamma} \tag{22}$$

Since under HE algorithm, as long as event $E$ happens, the final rank is correct, which finishes the proof. $\square$

Now, the following analysis will be condition on event $E$. First, we use $H^1$ to denote the original system that operates under HE algorithm. Next, we construct a coupled system $H^2$, under which the operator only verifies the feedback of item $I_j$ that has smallest number of verified samples (break tie arbitrarily), which means the system $H^2$ will stay idle even there are other feedback to be verified if the feedback queue for $I_j$ is empty. Also, under $H^2$, the operator only eliminate items if all items in set $B$ have identical number of verified samples.

**Lemma 3.** *A sufficient condition for system $H^2$ to finalize the rank is that the system verifies $\sum_{j=1}^{K} \frac{\lceil 16\gamma \log(T) \rceil}{\min\{\Delta_{j-1,j}, \Delta_{j,j+1}\}}$*

*Proof.* For convenience, we use $M_j$ to denote the quantity $\frac{\lceil 16\gamma \log(T) \rceil}{\min\{\Delta_{j-1,j}, \Delta_{j,j+1}\}}$. We know that under event $E$, when

$$\hat{\beta}_j(t) + \sqrt{\frac{\gamma \log(T)}{m_j(t)}} \leq \hat{\beta}_{j-1}(t) - \sqrt{\frac{\gamma \log(T)}{m_{j-1}(t)}} \tag{23}$$

or equivalently,

$$2\sqrt{\frac{\gamma \log(T)}{m_j(t)}} \leq \hat{\beta}_{j-1}(t) - \hat{\beta}_j(t) \tag{24}$$

The system $H^2$ can finalize the relative rank between $I_j$ and $I_i$ for all $i < j$. The equivalence for the second equation is because under $H^2$, elimination only occurs when all items in set $B$ have identical number of verified samples. Furthermore, under event $E$, a sufficient condition for elimination is

$$4\sqrt{\frac{\gamma \log(T)}{m_j(t)}} \leq \beta_{j-1} - \beta_j \tag{25}$$

This is because even under worst case where,

$$\beta_j = \hat{\beta}_j(t) + \sqrt{\frac{\gamma \log(T)}{m_j(t)}} \tag{26}$$

$$\beta_{j-1} = \hat{\beta}_{j-1}(t) - \sqrt{\frac{\gamma \log(T)}{m_{j-1}(t)}} \tag{27}$$

it suffices to distinguish both items. Similar arguments holds for pair $(I_j, I_{j+1})$, thus, in order to finalize the rank for item $I_j$, it suffices to have $M_j$ samples for all $j$.

Thus, it remains to show that the system will never verify items with $m_j(t) \geq M_j$. If $m_j(t) \geq M_j$, the set $I_j$ lies in the HE algorithm is a singleton set, and we will never verify the feedback of $I_j$. Therefore, the system finalized the rank before it verifies $\sum_{j=1}^K M_j$ feedback. $\qquad \square$

By the previous lemma, we only need to upper bound the expected time for system $H^2$, by our design, the system only verify samples with least number of verified samples. Thus, the expected time that system $H^2$ verifies desired number of feedback is upper bounded by a $M/M/1$ queuing system with arrival rate $\alpha_1$ and service rate $\mu$.

**Lemma 4** (Pathwise comparison of elimination completion times of $H^1$ and $H^2$). *Fix an arbitrary realization $\omega$ of all primitive randomness in the system (arrival times, choices, feedback values, and service times).[1] Let $H^1$ be the original system that runs the HE policy, and let $H^2$ be the auxiliary system defined as follows:*

- *$H^2$ observes exactly the same verified feedback samples as $H^1$ on the path $\omega$. In particular, for each item $k$ and time $t$ we have the same number of verified samples $m_k(t)$ and the same empirical mean $\hat{\beta}_k(t)$ as in $H^1$.*

- *$H^2$ therefore uses the same confidence bounds*

$$LCB_k(t) = \hat{\beta}_k(t) - \sqrt{\frac{\gamma \log T}{m_k(t)}}, \qquad UCB_k(t) = \hat{\beta}_k(t) + \sqrt{\frac{\gamma \log T}{m_k(t)}}.$$

- *The only difference between $H^1$ and $H^2$ is the timing of the eliminations: $H^1$ applies the hierarchical elimination rule as soon as it is satisfied, whereas $H^2$ is more conservative and is allowed to update the order sets only at certain "synchronization" times (e.g., when all items in $\mathcal{B}(t)$ have the same number of verified samples). Thus $H^2$ may delay an elimination that $H^1$ would already perform, but it never uses more information (samples) than $H^1$ at any time.*

*Let $\tau^{(1)}(\omega)$ and $\tau^{(2)}(\omega)$ denote the (random) times at which all eliminations are completed (i.e., all order sets $\mathcal{A}^q$ are singletons) in $H^1$ and $H^2$, respectively, on the sample path $\omega$. Then*

$$\tau^{(1)}(\omega) \leq \tau^{(2)}(\omega) \qquad \text{for every realization } \omega.$$

---

[1]That is, $\omega$ fixes the entire sequence of feedback observations that would be obtained whenever a particular feedback is selected for verification.

*Proof.* Fix an arbitrary realization $\omega$ of the primitive randomness throughout the proof.

For each item $k$, the empirical mean $\hat{\beta}_k(t)$ is computed from the $m_k(t)$ verified samples observed up to time $t$:

$$\hat{\beta}_k(t) \;=\; \frac{m_k^p(t)}{m_k(t)},$$

and the confidence bounds in both $H^1$ and $H^2$ are

$$LCB_k(t) \;=\; \hat{\beta}_k(t) - \sqrt{\frac{\gamma \log T}{m_k(t)}}, \qquad UCB_k(t) \;=\; \hat{\beta}_k(t) + \sqrt{\frac{\gamma \log T}{m_k(t)}}.$$

Along the fixed path $\omega$, the count $m_k(t)$ is nondecreasing in $t$. Hence $UCB_k(t)$ is nonincreasing and $LCB_k(t)$ is nondecreasing in $t$.

Therefore, for any two times $t_1 \leq t_2$ and any items $i, j$,

$$UCB_i(t_2) \;\leq\; UCB_i(t_1), \qquad LCB_j(t_2) \;\geq\; LCB_j(t_1). \tag{28}$$

In particular, if at some time $t_1$ we have $UCB_i(t_1) < LCB_j(t_1)$, then for all $t_2 \geq t_1$ we also have $UCB_i(t_2) < LCB_j(t_2)$.

Fix an item $k \in \{1, \ldots, K\}$ which is eventually assigned a final position by the hierarchical elimination rule (this happens for all items on the path $\omega$).

*Definition of $t_k^*(\omega)$.* Consider the process $H^1$ on the path $\omega$. For item $k$, define

$$t_k^*(\omega) \;:=\; \inf\Big\{ t \geq 0 \,:\, \exists\, j, q \text{ such that } I_k, I_j \in \mathcal{A}^q(t),\; UCB_k(t) < LCB_j(t) \Big\}.$$

Thus $t_k^*(\omega)$ is the *earliest* time at which there exists some item $j$ in the same order set $\mathcal{A}^q$ as $k$ such that $UCB_k < LCB_j$, i.e., the earliest time at which $k$ becomes *eliminable* according to the HE rule.

By construction of $H^1$, the algorithm eliminates $I_k$ *as soon as* the elimination condition is satisfied. Therefore, on the path $\omega$ we have

$$\tau_k^{(1)}(\omega) \;\leq\; t_k^*(\omega), \tag{29}$$

where $\tau_k^{(1)}(\omega)$ is the (random) time at which $I_k$ is moved out of $\mathcal{A}^q$ into a lower level (or becomes a singleton) in $H^1$.

*Behavior of $H^2$.* In $H^2$ we use the *same* empirical means and confidence bounds as in $H^1$, because $H^2$ is defined on top of the same verification trajectory: for each $t$ and each item $k$,

$$\hat{\beta}_k^{(2)}(t) = \hat{\beta}_k^{(1)}(t), \quad LCB_k^{(2)}(t) = LCB_k^{(1)}(t), \quad UCB_k^{(2)}(t) = UCB_k^{(1)}(t).$$

The only difference is that $H^2$ is allowed to update the order sets (which include moving $I_k$ to lower levels) only at a subsequence of times $\{t_r\}_{r \geq 1}$ (the "synchronization times") which are nondecreasing and satisfy $t_r \to \infty$ as $r \to \infty$. For concreteness, one may think of $t_r$ as the first time at which every item in $\mathcal{B}(t)$ has received at least $r$ verified samples, but the argument below only uses the fact that

$$t_1 \leq t_2 \leq \ldots, \quad t_r \uparrow \infty.$$

Let $\tau_k^{(2)}(\omega)$ be the time at which $I_k$ is eliminated in $H^2$. By definition of the algorithm $H^2$, there must exist an index $r_k$ such that

$$\tau_k^{(2)}(\omega) = t_{r_k},$$

and at time $t_{r_k}$ we have

$$\exists\, j, q \text{ with } I_k, I_j \in \mathcal{A}^q(t_{r_k}) \quad \text{and} \quad UCB_k(t_{r_k}) < LCB_j(t_{r_k}),$$

otherwise $H^2$ would not eliminate $I_k$ at time $t_{r_k}$.

Since $t_k^*(\omega)$ is the *earliest* time when such a pair $(k, j)$ exists, we must have

$$t_k^*(\omega) \;\leq\; t_{r_k} = \tau_k^{(2)}(\omega). \tag{30}$$

Indeed, the condition "there exists $j$ with $UCB_k < LCB_j$ in the same order set as $k$" is already satisfied at time $t_k^*(\omega)$ by definition of $t_k^*$, and by the monotonicity in equation 28 it continues to hold for all $t \geq t_k^*(\omega)$, including $t_{r_k}$.

Combining equation 29 and equation 30, we obtain

$$\tau_k^{(1)}(\omega) \ \leq \ t_k^*(\omega) \ \leq \ \tau_k^{(2)}(\omega), \qquad \forall k \in \{1, \ldots, K\}.$$

The elimination completion time in system $H^i$ $(i = 1, 2)$ on the path $\omega$ is the first time at which every item has been assigned its final level, i.e.,

$$\tau^{(i)}(\omega) \ = \ \max_{k=1,\ldots,K} \tau_k^{(i)}(\omega).$$

Using the item-wise inequality derived above, we conclude

$$\tau^{(1)}(\omega) \ = \ \max_k \tau_k^{(1)}(\omega) \ \leq \ \max_k \tau_k^{(2)}(\omega) \ = \ \tau^{(2)}(\omega).$$

Since $\omega$ was arbitrary, this holds for every realization of the primitive randomness, which completes the proof. $\square$

**Lemma 5.** *For a $M/M/1$ queue, the expected number of departures by time $t$ with arrival rate $\alpha_1$ and service rate $\mu$, for $\alpha_1 > \mu$:*

$$\mathbb{E}[D(t)] \geq \mu t - \frac{1}{\alpha_1} - o(1) \tag{31}$$

We require

$$t \geq \frac{1}{\alpha_1} + \frac{\sum_{j=1}^K M_j}{\mu} + o(1) \tag{32}$$

Therefore, since $M_T = O(T)$, if $\gamma > 0.5$, the regret is upper bounded by

$$Reg(T) \leq O\Big(\sum_{k=1}^K \frac{\log(T)\Delta}{\min\{\Delta_{k-1,k}, \Delta_{k-1,k}\}^2 \mu} + \frac{\Delta}{\alpha_1}\Big) \tag{33}$$

## A.2 PROOF OF THEOREM 3

In bi-criteria setting, we need to bound the expected time that either one of the criteria is met, and by the convexity of minimum function, it suffices to analyze the second criteria in order to derive an upper bound.

**Lemma 6.** *With high probability, if $I_j \in \Psi$, the true mean $\beta_j$ is in $[L\tilde{C}B_j(t), U\bar{C}B_j(t)]$ for any $t$.*

*Proof.* Define $\bar{q}_j(t)$ to be the average $q_j$ of all arrivals up to time $t$. We have

$$\mathbb{P}\left(\beta_j < L\tilde{C}B_j(t)\right) = \mathbb{P}\left(\beta_j < \frac{n_j^p(t) - n_j(t)\bar{\phi}}{n_j(t)} - \sqrt{\frac{\gamma \log(T)}{m_j(t)}}\right) \tag{34}$$

$$= \mathbb{P}\left(\frac{n_j^p(t)}{n_j(t)} - \bar{\phi} - \beta_j > \sqrt{\frac{\gamma \log(T)}{m_j(t)}}\right) \tag{35}$$

$$= \mathbb{P}\left(\frac{n_j^p(t)}{n_j(t)} - (\phi_j \bar{q}_j(t) + (1-\phi_j)\beta_j) + \underbrace{(\phi_j \bar{q}_j(t) + (1-\phi_j)\beta_j) - \bar{\phi} - \beta_j}_{\leq 0} > \sqrt{\frac{\gamma \log(T)}{m_j(t)}}\right) \tag{36}$$

$$\leq \mathbb{P}\left(\frac{n_j^p(t)}{n_j(t)} - (\phi_j \bar{q}_j(t) + (1-\phi_j)\beta_j) > \sqrt{\frac{\gamma \log(T)}{m_j(t)}}\right) \tag{37}$$

$$\leq \exp(-2\gamma \log(T)) \tag{38}$$

$$= T^{-2\gamma} \tag{39}$$

and

$$\mathbb{P}\left(\beta_j > U\bar{C}B_j(t)\right) = \mathbb{P}\left(\beta_j < \frac{n_j^p(t) - n_j(t)\bar{\phi}}{n_j(t)} + \sqrt{\frac{\gamma \log(T)}{m_j(t)}}\right) \tag{40}$$

$$= \mathbb{P}\left(-\frac{n_j^p(t)}{n_j(t)} - \bar{\phi} + \beta_j > \sqrt{\frac{\gamma \log(T)}{m_j(t)}}\right) \tag{41}$$

$$= \mathbb{P}\left(-\frac{n_j^p(t)}{n_j(t)} + (\phi_j \bar{q}_j(t) + (1-\phi_j)\beta_j) - \underbrace{(\phi_j \bar{q}_j(t) + (1-\phi_j)\beta_j) - \bar{\phi} + \beta_j}_{\leq 0} > \sqrt{\frac{\gamma \log(T)}{m_j(t)}}\right) \tag{42}$$

$$\leq \mathbb{P}\left(\left|\frac{n_j^p(t)}{n_j(t)} - (\phi_j \bar{q}_j(t) + (1-\phi_j)\beta_j)\right| > \sqrt{\frac{\gamma \log(T)}{m_j(t)}}\right) \tag{43}$$

$$\leq 2\exp(-2\gamma \log(T)) \tag{44}$$

$$= 2T^{-2\gamma} \tag{45}$$

$\square$

Similarly, by union bound, we have

$$\mathbb{P}\left(\beta_j \in [L\tilde{C}B_j(t), U\bar{C}B_j(t)], \text{ for any } t\right) \geq 1 - O(T^{1-2\gamma}) \tag{46}$$

Thus, we will condition on the above event for the following analysis. First, in order to finalized the rank, by similar arguments, it suffices to have $n_j(t) \geq N_j$, where

$$N_j := \frac{\lceil 16\gamma \log(T) \rceil}{\min\{\delta_{j-1,j}, \delta_{j,j+1}\}} \tag{47}$$

Finally, for items in $\Psi$, the expected marginal time contribution to the system is bounded by

$$\mathbb{E}\left[\inf_t \{n_j(t) \geq N_j \cup m_j(t) \geq M_j\}\right] \tag{48}$$

$$\leq \min\{\mathbb{E}[\inf_t \{n_j(t) \geq N_j\}], \mathbb{E}[\inf_t \{m_j(t) \geq M_j\}]\} \tag{49}$$

plug in the previous results, we finishes the proof of the following regret upper bound

$$Reg(T) \leq O\left(\sum_{k \in \Psi} \Delta \min\{\frac{\log(T)}{\min\{\Delta_{k-1,k}, \Delta_{k-1,k}\}^2 \mu}, \frac{\log(T)}{\min\{\delta_{k-1,k}^2, \delta_{k,k+1}^2\}\alpha_K}\} + \sum_{k \notin \Psi} \frac{\Delta \log(T)}{\min\{\Delta_{k-1,k}, \Delta_{k-1,k}\}^2 \mu} + \frac{\Delta}{\alpha_1}\right) \tag{50}$$

### A.3 PROOF OF THEOREM 2

For any arrival and service sequence with fixed customer choice, we can define the embedded sample space by:

$$\Omega := ([K]^{K+1} \times \{0,1\})^{N_T + M_T}, \tag{51}$$

where $M_T$ and $N_T$ are the total number of verifications and that of arrivals. The sample space is defined condition on an event sequence, where there are two types of events, arrival and verification completion. For arrival event, we use the triplet $(R_{t_i}, C_{t_i}, Y_{t_i})$ to denote the rank at time $t_i$, the customer choice at time $t_i$, and the realized feedback for this choice. Note that $Y_{t_i}$ is the superficial feedback of this arrival. For verification completion event, we use another triplet $(R_{t_i}, I_{t_i}, X_{t_i})$ to denote the rank at time $t_i$, the item whose feedback just being verified, and the value of true feedback.

Next, we define the history:

$$\mathcal{H}_{t_n} = ((R_{t_0}, C_{t_0}, Y_{t_0}), \ldots, (R_{t_n}, C_{t_n}/I_{t_n}, Y_{t_n}/X_{t_n})), \tag{52}$$

where the "/" means "or" accounting for the uncertainty of event type at time $t_n$. For a ranking policy $\pi^r$ and scheduling policy $\pi^s$, we have:

$$R_{t_n} = \pi^r_{t_n}(\mathcal{H}_{t_{n-1}}), \; I_{t_n} = \pi^s_{t_n}(\mathcal{H}_{t_{n-1}}) \tag{53}$$

Next, we define the probability measure $\mathbb{P}_\nu$ of the interconnection of policy and a fixed event sequence of the original instance $\nu$. Formally, for $\omega \in \Omega$, we have:

$$\mathbb{P}_\nu(\omega) = \prod_{i \in \mathcal{N}} \sum_{C_{t_i}=1}^{K} \mathbb{P}_{C_{t_i}}(Y_{t_i}) \mathbf{1}\{C_{t_i} = c(R_{t_i})\}$$

$$\prod_{i \in \mathcal{M}} \sum_{I_{t_i}=1}^{K} \mathbb{P}_{I_{t_i}}(Y_{t_i}) \mathbf{1}\{I_{t_i} = \pi^s_{t_i}(\mathcal{H}_{t_{i-1}})\}, \tag{54}$$

where $\mathbb{P}_{C_{t_i}}$ is Bernoulli distribution with mean $\phi_{C_{t_i}} q_{C_{t_i}}(t_i) + (1 - \phi_{C_{t_i}})\beta_{C_{t_i}}$, and $\mathbb{P}_{I_{t_i}}$ is Bernoulli distribution with mean $\beta_{I_{t_t}}$. The set $\mathcal{M}$ and $\mathcal{N}$ represent the index set for verification completion event and arrival event respectively.

We construct alternative instance $\nu^1$, where we enlarge the quality parameter for item $I_j$ to

$$\beta^1_j = \beta_{j-1} + \epsilon, \text{ for } \epsilon > 0 \tag{55}$$

One key setting is that $q_j(t)$ is unknown can be arbitrary selected for any time $t$ as long as $q_j(t) \in [0, 1]$. Thus, in general, the larger the $\phi_j$ is, the less information contained in the arrival event. Specifically, we will discuss case by case:

**Case 1:** consider when $\phi_j \geq \frac{\Delta_{j-1,j}}{\Delta_{j-1,j}+1}$, it is possible that set

$$q_j(t) - q^1_j(t) = \frac{1 - \phi_j}{\phi_j}(\Delta_{j-1,j} + \epsilon) \tag{56}$$

Consequently,

$$KL(\mathbb{P}_\nu || \mathbb{P}_{\nu^1}) = \mathbb{E}_\nu[m_j(T)] KL(Ber(\beta_j) || Ber(\beta_{j-1} + \epsilon)), \tag{57}$$

where

$$m_j(T) = \mathbb{E}_\nu\left[\sum_{i \in \mathcal{M}} \mathbf{1}\{I_{t_t} = I_j\}\right] \tag{58}$$

We define event

$$A = \{\text{At least on half of the events, the policy rank } I_j \text{ before } I_{j-1}\} \tag{59}$$

Further, since the inter-event time is stochastically lower bounded by a exponential random variable with mean $\frac{1}{1+\mu}$, as a result,

$$Reg(T) \geq \frac{M_T + N_T}{2(\mu + 1)} \Delta_{j-1,j} \mathbb{P}_\nu(A) \tag{60}$$

$$Reg(T)^1 \geq \frac{M_T + N_T}{2(\mu + 1)} \epsilon \mathbb{P}_{\nu^1}(A^c) \tag{61}$$

Thus,

$$Reg(T) + Reg(T)^1 \geq \frac{M_T + N_T}{2(\mu + 1)} \min\{\epsilon, \Delta_{j-1,j}\} \left[\mathbb{P}_\nu(A) + \mathbb{P}_{\nu^1}(A^c)\right] \tag{62}$$

$$\geq \frac{M_T + N_T}{4(\mu + 1)} \min\{\epsilon, \Delta_{j-1,j}\} e^{-KL(\mathbb{P}_\nu || \mathbb{P}_{\nu^1})} \tag{63}$$

$$= \frac{M_T + N_T}{4(\mu + 1)} \min\{\epsilon, \Delta_{j-1,j}\} e^{-\mathbb{E}_\nu[m_j(T)] KL(Ber(\beta_j) || Ber(\beta_{j-1} + \epsilon))} \tag{64}$$

Equivalently,

$$\frac{\mathbb{E}_\nu[m_j(T)]}{\log(T)} \geq \frac{1}{KL(Ber(\beta_j) || Ber(\beta_{j-1} + \epsilon))} \left[\frac{\log(M_T + N_T)}{\log(T)} + \frac{\log(\min\{\epsilon, \Delta_{j-1,j}\})}{4(\mu + 1)\log(T)} - \frac{\log(Reg(T) + Reg(T)^1)}{\log(T)}\right] \tag{65}$$

**Definition 1.** *For a consistent policy $\pi$, we require*

$$Reg(T) + Reg(T)^1 \leq C_\xi T^\xi, \text{ for any } \xi > 0 \tag{66}$$

Thus,

$$\limsup_{T\to\infty} \frac{\log(Reg(T) + Reg(T)^1)}{\log(T)} \leq \limsup_{T\to\infty} \frac{\xi \log(T) + \log(C_\xi)}{\log(T)} \tag{67}$$

take limit $\xi \to 0$:

$$\limsup_{T\to\infty} \frac{\log(Reg(T) + Reg(T)^1)}{\log(T)} = 0 \tag{68}$$

Consequently,

$$\liminf_{T\to\infty} \frac{\mathbb{E}_\nu[m_j(T)]}{\log(T)} \geq \liminf_{T\to\infty} \frac{1}{KL(Ber(\beta_j)||Ber(\beta_{j-1}+\epsilon))} \frac{\log(M_T + N_T)}{\log(T)} \tag{69}$$

$$\geq \liminf_{T\to\infty} \frac{1}{KL(Ber(\beta_j)||Ber(\beta_{j-1}+\epsilon))} \frac{\log(N_T)}{\log(T)} \tag{70}$$

We also know that $N_T$ is the total number of arrivals by time $T$, and by law of large numbers, we know

$$\liminf_{T\to\infty} \frac{\mathbb{E}_\nu[m_j(T)]}{\log(T)} \geq \frac{1}{KL(Ber(\beta_j)||Ber(\beta_{j-1}+\epsilon))} \tag{71}$$

Finally, we take the limit for $\epsilon \to 0$,

$$\liminf_{T\to\infty} \frac{\mathbb{E}_\nu[m_j(T)]}{\log(T)} \geq \frac{1}{KL(Ber(\beta_j)||Ber(\beta_{j-1}))} \tag{72}$$

As a result, the expected time for system to fulfill the above condition is

$$\Omega\left(\frac{\log(T)}{KL(Ber(\beta_j)||Ber(\beta_{j-1}))}\right) \tag{73}$$

**Case 2:** consider when $\phi_j < \frac{\Delta_{j-1,j}}{\Delta_{j-1,j}+1}$, it is impossible to have $q_j(t)$ and $q_j^1(t)$ by the above equation, which leads to the information gain for the arrival event. However, the it can still be:

$$(q_j(t), q_j^1(t)) = \arg\min\{KL(Ber(\phi_j q_j(t) + (1-\phi_j)\beta_j)||(Ber(\phi_j q_j^1(t) + (1-\phi_j)(\beta_{j-1}+\epsilon)))\} \tag{74}$$

For convenience, we denote

$$d_{j,j^1} := \inf_{q_j(t),q_j^1(t)} \{KL(Ber(\phi_j q_j(t) + (1-\phi_j)\beta_j)||(Ber(\phi_j q_j^1(t) + (1-\phi_j)(\beta_{j-1}+\epsilon)))\} \tag{75}$$

Thus,

$$KL(\mathbb{P}_\nu||\mathbb{P}_{\nu^1}) = \mathbb{E}_\nu[m_j(T)]KL(Ber(\beta_j)||Ber(\beta_{j-1}+\epsilon)) + \mathbb{E}_\nu[n_j(T)]d_{j,j^1} \tag{76}$$

By similar arguments, we have

$$\liminf_{T\to\infty} \frac{\mathbb{E}_\nu[m_j(T) + n_j(T)]}{\log(T)} \geq \frac{1}{\max\{d_{j,j-1}, KL(Ber(\beta_j)||Ber(\beta_{j-1}))\}} \tag{77}$$

And there for the expected time the system should spend is

$$\Omega\left(\frac{\log(T)}{(\mu+1)\max\{d_{j,j-1}, KL(Ber(\beta_j)||Ber(\beta_{j-1}))\}}\right) \tag{78}$$

Next, we construct instance $\nu^2$, where we set

$$\beta_j^2 = \beta_{j+1} - \epsilon, \text{ for } \epsilon > 0 \tag{79}$$

Follow similar arguments, we have:

**Case 1:** $\phi_j \geq \frac{\Delta_{j,j+1}}{\Delta_{j,j+1}+1}$, the expected time system should spend before the condition is met is

$$\Omega(\frac{\log(T)}{KL(Ber(\beta_j)||Ber(\beta_{j+1}))}) \tag{80}$$

**Case 2:** $\phi_j < \frac{\Delta_{j,j+1}}{\Delta_{j,j+1}+1}$, the expected time system should spend before the condition is met is

$$\Omega(\frac{\log(T)}{(\mu+1)\max\{d_{j,j^2}, KL(Ber(\beta_j)||Ber(\beta_{j+1}))\}}) \tag{81}$$

Define the sets:

$$\Gamma_1 = \{I_j : \phi_j \geq \frac{\Delta_{j,j+1}}{\Delta_{j,j+1}+1}, \phi_j \geq \frac{\Delta_{j-1,j}}{\Delta_{j-1,j}+1}\}, \tag{82}$$

$$\Gamma_2 = \{I_j : \phi_j < \frac{\Delta_{j,j+1}}{\Delta_{j,j+1}+1}, \phi_j < \frac{\Delta_{j-1,j}}{\Delta_{j-1,j}+1}\}, \tag{83}$$

$$\Gamma_3 = \{I_j : \phi_j \geq \frac{\Delta_{j,j+1}}{\Delta_{j,j+1}+1}, \phi_j < \frac{\Delta_{j-1,j}}{\Delta_{j-1,j}+1}\}, \tag{84}$$

$$\Gamma_4 = \{I_j : \phi_j \geq \frac{\Delta_{j,j+1}}{\Delta_{j,j+1}+1}, \phi_j < \frac{\Delta_{j-1,j}}{\Delta_{j-1,j}+1}\}. \tag{85}$$

For $I_j \in \Gamma_1$, the expected time system spend is

$$\Omega(\frac{\log(T)}{\mu\min\{KL(Ber(\beta_j)||Ber(\beta_{j-1})), KL(Ber(\beta_j)||Ber(\beta_{j+1}))\}}) \tag{86}$$

For $I_j \in \Gamma_2$, the expected time system spend is

$$\Omega(\frac{\log(T)}{(\mu+1)\min\{\max\{d_{j,j^2}, KL(Ber(\beta_j)||Ber(\beta_{j+1}))\}, \max\{d_{j,j^1 1}, KL(Ber(\beta_j)||Ber(\beta_{j-1}))\}\}}) \tag{87}$$

For $I_j \in \Gamma_3$, the expected time system spend is

$$\Omega(\max\{\frac{\log(T)}{(\mu+1)\max\{d_{j,j^1}, KL(Ber(\beta_j)||Ber(\beta_{j-1}))\}}, \frac{\log(T)}{KL(Ber(\beta_j)||Ber(\beta_{j+1}))}\}) \tag{88}$$

For $I_j \in \Gamma_4$, the expected time system spend is

$$\Omega(\max\{\frac{\log(T)}{(\mu+1)\max\{d_{j,j^2}, KL(Ber(\beta_j)||Ber(\beta_{j+1}))\}}, \frac{\log(T)}{KL(Ber(\beta_j)||Ber(\beta_{j-1}))}\}) \tag{89}$$

And, we define:

$$C_j^1(\mu) = \frac{1}{\mu\min\{KL(Ber(\beta_j)||Ber(\beta_{j-1})), KL(Ber(\beta_j)||Ber(\beta_{j+1}))\}}, \tag{90}$$

$$C_j^2(\mu) = \frac{1}{(\mu+1)\min\{\max\{d_{j,j^2}, KL(Ber(\beta_j)||Ber(\beta_{j+1}))\}, \max\{d_{j,j^1 1}, KL(Ber(\beta_j)||Ber(\beta_{j-1}))\}\}}, \tag{91}$$

$$C_j^3(\mu) = \max\{\frac{1}{(\mu+1)\max\{d_{j,j^1}, KL(Ber(\beta_j)||Ber(\beta_{j-1}))\}}, \frac{1}{KL(Ber(\beta_j)||Ber(\beta_{j+1}))}\}, \tag{92}$$

$$C_j^4(\mu) = \max\{\frac{1}{(\mu+1)\max\{d_{j,j^2}, KL(Ber(\beta_j)||Ber(\beta_{j+1}))\}}, \frac{1}{KL(Ber(\beta_j)||Ber(\beta_{j-1}))}\}. \tag{93}$$

Lastly, before the conditions for all items are met, the system will incur a polynomial regret with rate at least $\Delta_{min}$, therefore, we have

$$\liminf_{T\to\infty} \frac{Reg(T)}{\log(T)} \geq \Omega(\Delta_{min}\sum_{\xi=1}^{4}\sum_{k=1}^{K}\mathbf{1}\{I_k \in \Gamma_\xi\}C_k^\xi(\mu)), \tag{94}$$

A.4 PROOF OF THEOREM 4

We first prove the lemma 1.

*Proof.* For any verifier $V_i$, we know that if no elimination occurs, the minimum deficit is upper bounded by the maximum of $M_{i,T}$ exponential random variables with mean $\frac{1}{\mu_{min}}$, where $M_{i,T}$ is the total number of verifications completed by verifier $V_i$. This is because the total deficits for any verifier $V_i$ is always zero, and thus the maximum deficit within the verifier is at most the absolute value of the minimum deficit. While the minimum deficit are driven by the maximum service time. Thus,

$$\limsup_{t\to\infty} \mathbb{P}\left(\frac{\theta_{ij}(t)}{t} > \epsilon\right) \leq \limsup_{t\to\infty} \mathbb{P}\left(\max\{Z_1,\ldots,Z_{M_{j,t}}\} > \epsilon\right) \tag{95}$$

$$\leq \limsup_{t\to\infty} 1 - (1 - e^{-\mu_{min}\epsilon t})^{M_{j,t}} \tag{96}$$

$$\leq \limsup_{t\to\infty} 1 - (1 - e^{-\mu_{min}\epsilon t})^{O(t)} \tag{97}$$

$$= 0 \tag{98}$$

holds for any $\epsilon > 0$, which finishes the proof of asymptotic results. For finite time analysis, we only need to find the upper bound of $\mathbb{E}\left[\frac{\theta_{ij}(t)}{t}\right]$, we derive the results as follows:

$$\mathbb{E}\left[\frac{\theta_{ij}(t)}{t}\right] \leq \sum_{m=1}^{\infty} \mathbb{P}(M_t = m) \int_0^\infty \mathbb{P}\left(\frac{\theta_{ij}(t)}{t} > \epsilon\right) d\epsilon \tag{99}$$

$$\leq \sum_{m=1}^{\infty} \mathbb{P}(M_t = m) \int_0^\infty \mathbb{P}\left(\max\{Z_1,\ldots,Z_m\} > \epsilon\right) d\epsilon \tag{100}$$

$$= \sum_{m=1}^{\infty} \mathbb{P}(M_t = m) \int_0^\infty 1 - (1 - e^{-\mu_{min}\epsilon t})^m d\epsilon \tag{101}$$

We aim to evaluate the integral:

$$I = \int_0^\infty \left[1 - \left(1 - e^{-\mu_{\min}\epsilon t}\right)^m\right] d\epsilon$$

Let us perform a substitution to non-dimensionalize the integral:

$$x = \mu_{\min} t\epsilon \quad \Rightarrow \quad \epsilon = \frac{x}{\mu_{\min} t}, \quad d\epsilon = \frac{dx}{\mu_{\min} t}$$

Substituting these into the integral $I$:

$$I = \int_0^\infty \left[1 - \left(1 - e^{-x}\right)^m\right] \frac{dx}{\mu_{\min} t} = \frac{1}{\mu_{\min} t} \int_0^\infty \left[1 - \left(1 - e^{-x}\right)^m\right] dx$$

Let us denote the dimensionless integral as $J$:

$$J = \int_0^\infty \left[1 - \left(1 - e^{-x}\right)^m\right] dx$$

Thus,

$$I = \frac{J}{\mu_{\min} t}$$

We can expand the term $(1 - e^{-x})^m$ using the binomial theorem:

$$\left(1 - e^{-x}\right)^m = \sum_{k=0}^{m} \binom{m}{k}(-1)^k e^{-kx}$$

Therefore, the integrand becomes:

$$1 - \left(1 - e^{-x}\right)^m = 1 - \sum_{k=0}^{m} \binom{m}{k}(-1)^k e^{-kx} = \sum_{k=1}^{m} \binom{m}{k}(-1)^{k+1} e^{-kx}$$

Substituting the expanded form into $J$:

$$J = \int_0^{\infty} \sum_{k=1}^{m} \binom{m}{k}(-1)^{k+1} e^{-kx} dx$$

Assuming uniform convergence (which holds here due to absolute convergence for each $x$), we can interchange the summation and integration:

$$J = \sum_{k=1}^{m} \binom{m}{k}(-1)^{k+1} \int_0^{\infty} e^{-kx} dx$$

The integral of the exponential function is straightforward:

$$\int_0^{\infty} e^{-kx} dx = \left[-\frac{1}{k}e^{-kx}\right]_0^{\infty} = \frac{1}{k}$$

Thus, $J$ simplifies to:

$$J = \sum_{k=1}^{m} \binom{m}{k}(-1)^{k+1}\frac{1}{k}$$

The summation:

$$\sum_{k=1}^{m} \binom{m}{k}\frac{(-1)^{k+1}}{k}$$

is known to equal the $m$-th **harmonic number**, denoted $H_m$, where:

$$H_m = \sum_{k=1}^{m} \frac{1}{k}$$

This can be verified for small values of $m$:

- For $m = 1$:
$$\sum_{k=1}^{1} \binom{1}{1}\frac{(-1)^{1+1}}{1} = 1 \cdot \frac{1}{1} = 1 = H_1$$

- For $m = 2$:
$$\sum_{k=1}^{2} \binom{2}{k}\frac{(-1)^{k+1}}{k} = 2 \cdot \frac{1}{1} - 1 \cdot \frac{1}{2} = \frac{3}{2} = H_2$$

- For $m = 3$:

$$\sum_{k=1}^{3} \binom{3}{k} \frac{(-1)^{k+1}}{k} = 3 \cdot \frac{1}{1} - 3 \cdot \frac{1}{2} + 1 \cdot \frac{1}{3} = \frac{11}{6} = H_3$$

Thus, in general:

$$J = H_m$$

Substituting back into the expression for $I$:

$$I = \frac{J}{\mu_{\min}t} = \frac{H_m}{\mu_{\min}t}$$

Therefore, the integral evaluates to the $m$-th harmonic number divided by $\mu_{\min}t$:

$$\int_0^{\infty} \left[ 1 - \left( 1 - e^{-\mu_{\min}\epsilon t} \right)^m \right] d\epsilon = \frac{H_m}{\mu_{\min}t}$$

where the harmonic number $H_m$ is defined as:

$$H_m = \sum_{k=1}^{m} \frac{1}{k}$$

As a result,

$$\mathbb{E}\left[ \frac{\theta_{ij}(t)}{t} \right] \leq \sum_{m=1}^{\infty} \mathbb{P}(M_t = m) \frac{H_m}{\mu_{\min}t} \tag{102}$$

$$\leq \sum_{m=1}^{\infty} \frac{e^{-\mu_{\max}t}(\mu_{\max}t)^m}{m!} \frac{H_m}{\mu_{\min}t} \tag{103}$$

We aim to evaluate the sum:

$$S = \sum_{m=1}^{\infty} \frac{e^{-\mu_{\max}t}(\mu_{\max}t)^m}{m!} \cdot \frac{H_m}{\mu_{\min}t}$$

where $H_m$ is the $m$-th harmonic number defined by:

$$H_m = \sum_{k=1}^{m} \frac{1}{k} = \rho + \psi(m+1)$$

with $\rho$ representing the Euler-Mascheroni constant and $\psi$ the digamma function.

Factor out the constants from the summation:

$$S = \frac{e^{-\mu_{\max}t}}{\mu_{\min}t} \sum_{m=1}^{\infty} \frac{(\mu_{\max}t)^m}{m!} H_m$$

Let $x = \mu_{\max}t$, then:

$$S = \frac{e^{-x}}{\mu_{\min}t} \sum_{m=1}^{\infty} \frac{x^m}{m!} H_m$$

The series to evaluate is:

$$\sum_{m=1}^{\infty} \frac{x^m}{m!} H_m$$

Using the definition $H_m = \rho + \psi(m+1)$, we have:

$$\sum_{m=1}^{\infty} \frac{x^m}{m!} H_m = \rho \sum_{m=1}^{\infty} \frac{x^m}{m!} + \sum_{m=1}^{\infty} \frac{x^m}{m!} \psi(m+1)$$

$$\rho \sum_{m=1}^{\infty} \frac{x^m}{m!} = \rho \left( e^x - 1 \right)$$

Express $\psi(m+1)$ using its integral representation:

$$\psi(m+1) = -\rho + \int_0^1 \frac{1 - t^m}{1 - t} dt$$

Substituting into the sum:

$$\sum_{m=1}^{\infty} \frac{x^m}{m!} \psi(m+1) = \sum_{m=1}^{\infty} \frac{x^m}{m!} \left( -\rho + \int_0^1 \frac{1 - t^m}{1 - t} dt \right)$$

Simplifying:

$$= -\rho \sum_{m=1}^{\infty} \frac{x^m}{m!} + \int_0^1 \frac{1}{1 - t} \sum_{m=1}^{\infty} \frac{(x(1 - t))^m}{m!} dt$$

Recognize the exponential series:

$$\sum_{m=1}^{\infty} \frac{(x(1 - t))^m}{m!} = e^{x(1-t)} - 1$$

Thus:

$$\sum_{m=1}^{\infty} \frac{x^m}{m!} \psi(m+1) = -\rho(e^x - 1) + \int_0^1 \frac{e^{x(1-t)} - 1}{1 - t} dt$$

Make a substitution $s = 1 - t$ $(ds = -dt)$:

$$= -\rho(e^x - 1) + \int_0^1 \frac{e^{xs} - 1}{s} ds$$

The integral is related to the exponential integral function $\text{Ei}(-x)$:

$$\int_0^1 \frac{e^{xs} - 1}{s} ds = \rho + \ln x + \text{Ei}(-x)$$

Therefore:

$$\sum_{m=1}^{\infty} \frac{x^m}{m!} H_m = \rho(e^x - 1) + (\rho + \ln x + \text{Ei}(-x)) e^x - \rho e^x = e^x \left( \rho + \ln x + \text{Ei}(-x) \right)$$

Substituting back into the expression for $S$:

$$S = \frac{e^{-x}}{\mu_{\min}t} \cdot e^x \left(\rho + \ln x + \text{Ei}(-x)\right) = \frac{\rho + \ln x + \text{Ei}(-x)}{\mu_{\min}t}$$

Recalling that $x = \mu_{\max}t$, we substitute:

$$S = \frac{\rho + \ln(\mu_{\max}t) + \text{Ei}(-\mu_{\max}t)}{\mu_{\min}t}$$

Thus, the sum evaluates to:

$$\sum_{m=1}^{\infty} \frac{e^{-\mu_{\max}t}(\mu_{\max}t)^m}{m!} \cdot \frac{H_m}{\mu_{\min}t} = \frac{\rho + \ln(\mu_{\max}t) + \text{Ei}(-\mu_{\max}t)}{\mu_{\min}t}$$

where Euler-Mascheroni Constant ($\rho$) is Approximately 0.5772, it is defined as the limiting difference between the harmonic series and the natural logarithm, and exponential Integral Function ($\text{Ei}(-x)$) defined for $x > 0$ is by:

$$\text{Ei}(-x) = -\int_x^{\infty} \frac{e^{-t}}{t}dt$$

This finishes the proof □

Now, for the regret upper bound, by 1, we know that the idle time for the system is $O(1)$. Also, we know that the sum of total deficits for each $B(t)$ are stochastically bounded by the maximum of $M_T$ exponential random variables, whose mean are at most $\mu_{\min}$. Finally, using the same arguments for system $H^2$, the upper bound holds.

## B  ADDITIONAL EXPERIMENTS

**Verifier Departure Rates.** Figures 5(a) and 5(b) illustrate the departure rates for verifiers $V_1$ and $V_2$, respectively. Verifier $V_1$ predominantly verifies item $I_1$ due to its higher verification rate for this item, with a smaller proportion allocated to verifying $I_2$ and none for $I_3$. Conversely, verifier $V_2$ focuses on verifying item $I_3$, followed by $I_2$, and does not verify $I_1$ given its low verification rate for this item.

**Convergence of Deficits.** To validate the convergence properties of our scheduling policy, we conduct additional experiments. Figure 3(a) shows that the deficits converge rapidly, stabilizing around $t \approx 70$ for a two-item, two-verifier system. Figure 3(b) demonstrates that deficits continue to converge efficiently even in larger systems with fifty items and ten verifiers.

**Robustness Experiments.** We performed additional experiments to show the robustness of our algorithm (adding noise for actual rates) if those assumptions are violated (with mean results reported).

It is noticeable that the misspecification of arrival rate affects little of the regret since we did not use it as an input. However, the misspecification of verification rate will affect the regret since we get suboptimal solution of equation (12). But the misspecification of distribution will not affect too much of the regret even for uniform one.

## C  REAL WORLD EXAMPLE

**Meituan Platform.** We provide an example from Meituan, a major Chinese food delivery and local services platform, to illustrate real-world human verification systems. Platforms like Meituan have implemented large-scale human verification to handle questionable feedback, which aligns with our theoretical framework.

Table 1: Robustness Analysis: Extra Regret Accumulated Under Different Misspecifications

| Extra Regret Accumulated | Misspec. of arrival rate (±0%) | Misspec. of arrival rate (±20%) | Misspec. of arrival rate (±50%) | Misspec. of arrival rate (±100%) | Misspec. of arrival process (Truncated Gaussian, same mean) | Misspec. of arrival process (Uniform, same mean) |
|---|---|---|---|---|---|---|
| Misspecification of verification rate (±0%) | 0.00% | 0.15% | -0.66% | 0.89% | 0.13% | 1.02% |
| Misspecification of verification rate (±20%) | 15.32% | 16.37% | 14.95% | 15.03% | N/A | N/A |
| Misspecification of verification rate (±50%) | 25.32% | 23.38% | 24.57% | 29.01% | N/A | N/A |
| Misspecification of verification process (Truncated Gaussian, same mean) | 1.89% | N/A | N/A | N/A | 2.03% | 6.20% |
| Misspecification of verification process (Uniform, same mean) | 11.96% | N/A | N/A | N/A | 15.11% | 28.92% |

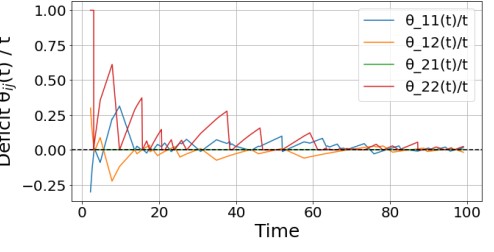

(a) Convergence on a 2-Item, 2-Verifier Instance

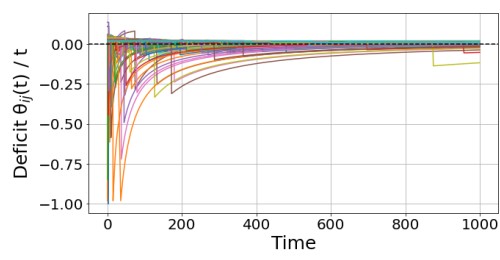

(b) Convergence on a 50-Item, 10-Verifier Instance

Figure 3: Convergence of Deficits

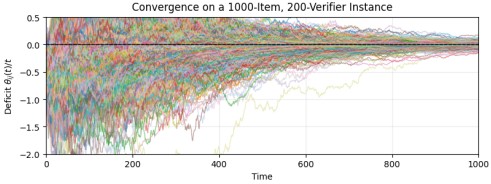

(a) Convergence on a 1000-Item, 200-Verifier Instance

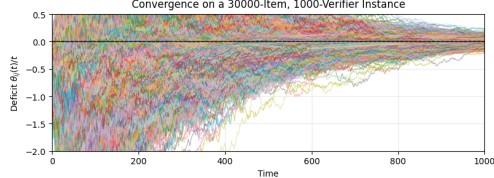

(b) Convergence on a 30000-Item, 1000-Verifier Instance

Figure 4: Convergence of Deficits

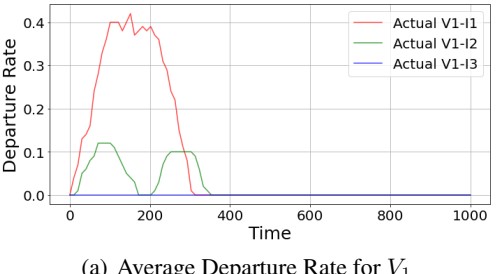

(a) Average Departure Rate for $V_1$

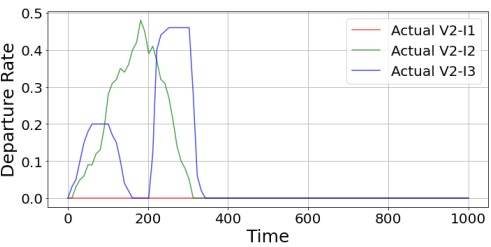

(b) Average Departure Rate for $V_2$

Figure 5: Multi-Server Experiments

Meituan's verification system addresses issues such as businesses disputing negative reviews and competitors alleging artificial review manipulation. Their "Xiaomei Review Panel" involves community members who vote on review disputes, creating a natural queueing system where verification requests exceed processing capacity.

The platform maintains neutrality by using independent reviewers selected based on activity level, registration duration, and demographic factors. Reviewers must maintain objectivity and follow strict confidentiality rules. The review process involves evidence submission, task assignment, anonymous voting, and majority-rule decisions.

This real-world implementation demonstrates the practical relevance of our theoretical model, where the relationship between regret bounds and verification efficiency $\mu$ becomes crucial for system performance.

## D  ALGORITHMS

---

**Algorithm 1** Hierarchical Elimination

**Input:** sets $\mathcal{A}^q(t)$ for $q \in [K]$, current time $t$, termination time $T$, verified sample size $m_k(t)$, total sample size $n_k(t)$

**for** $k = 1$ **to** $K$ **do**

$\quad LCB_k(t) = \hat{\beta}_k(t) - \sqrt{\frac{\gamma \log(T)}{m_k(t)}}$

$\quad UCB_k(t) = \hat{\beta}_k(t) + \sqrt{\frac{\gamma \log(T)}{m_k(t)}}$

**end for**

**for** $q = 1$ **to** $K$ **do**

$\quad$**for** $(i, j) \in \mathcal{A}^q$ **do**

$\quad\quad$**if** $UCB_i(t) < LCB_j(t)$ **then**

$\quad\quad\quad \mathcal{A}^q(t^+) = \mathcal{A}^q(t) \setminus \{I_i\}$

$\quad\quad\quad \mathcal{A}^{q+1}(t^+) = \mathcal{A}^{q+1}(t) \cup \{I_i\}$

$\quad\quad$**end if**

$\quad$**end for**

**end for**

HERank($\{\mathcal{A}^q(t^+)\}_{q=1}^{K}$)

---

**Algorithm 2** HERank

**Input:** sets $\mathcal{A}^q(t)$ for $q \in [K]$, $\mathcal{B} = \emptyset$

**for** $q = 1$ **to** $K$ **do**

$\quad$**if** $|\mathcal{A}^q| > 1$ **then**

$\quad\quad \mathcal{B} = \mathcal{B} \cup \mathcal{A}^q$

$\quad$**end if**

**end for**

**for** $(p, q)$ in $[K]^2$ **do**

$\quad$**if** $|\mathcal{A}^p| \leq 1$ and $|\mathcal{A}^q| \leq 1$ **then**

$\quad\quad$**if** $p < q$ **then**

$\quad\quad\quad$ Rank $\mathcal{A}^p$ before $\mathcal{A}^q$

$\quad\quad$**else**

$\quad\quad\quad$ Rank $\mathcal{A}^q$ before $\mathcal{A}^p$

$\quad\quad$**end if**

$\quad$**end if**

**end for**

**for** $I_k$ in $\mathcal{B}$ **do**

$\quad$ Rank in ascending order according to $n_k(t)$, use smaller $n_k(t) - m_k(t)$ for tie breaking

**end for**

Rank $\mathcal{B}$ before other items

---

---

**Algorithm 3** Deficit Hierarchical Elimination (DHE) Scheduling Policy

---

**Input:**
    Item set $\mathcal{I} = \{I_1, \ldots, I_K\}$
    Verifier set $\mathcal{V} = \{V_1, \ldots, V_N\}$
    Verification rates $\mu_{ij}$ for verifier $V_i$ and item $I_j$
    Time horizon $T$
    HE ranking policy that maintains order sets $\{\mathcal{A}^q\}_{q=1}^K$

**State variables:**
    $\mathcal{B}(t)$: union of non-singleton order sets under HE ranking at time $t$
    $Q_j(t)$: queue length of feedback for item $I_j$
    $S_{ij}(t)$: total service time spent by verifier $V_i$ on item $I_j$ up to time $t$
    $\theta_{ij}(t)$: deficit of pair $(i, j)$ at time $t$

**Procedure Initialize_DHE$(\mathcal{B})$:**
    *// Solve fair allocation LP for current ambiguous set $\mathcal{B}$*
    Solve
        $\max_{x_{ij}} \ \min_{j \in \mathcal{B}} \sum_{i=1}^N x_{ij}\mu_{ij}$
        subject to $\sum_{j \in \mathcal{B}} x_{ij} \leq 1$ for all $i$, and $x_{ij} \geq 0$
    Obtain optimal solution $x_{ij}^*(\mathcal{B})$ and optimal value $z^*(\mathcal{B})$
    *// Reset service times and deficits (local time origin for this $\mathcal{B}$)*
    **for** each verifier $i = 1, \ldots, N$ **do**
        **for** each item $j = 1, \ldots, K$ **do**
            $S_{ij} \leftarrow 0$
            $\theta_{ij} \leftarrow 0$
        **end for**
    **end for**
    Return $x_{ij}^*(\mathcal{B})$

**Main loop (event-driven, $t$ from $0$ to $T$):**
Initialize HE ranking; compute initial $\mathcal{B}(0)$
$x_{ij}^* \leftarrow$ Initialize_DHE$(\mathcal{B}(0))$
Set $t \leftarrow 0$
**while** $t \leq T$ **do**
    Advance $t$ to next event time $t^+$ (arrival or verification completion)
    $t \leftarrow t^+$
    **if** HE ranking eliminates some items and changes $\{\mathcal{A}^q\}$ **then**
        Update $\mathcal{B}(t)$ as union of non-singleton sets $\mathcal{A}^q$
        $x_{ij}^* \leftarrow$ Initialize_DHE$(\mathcal{B}(t))$
    **end if**
    **if** a verification by verifier $V_i$ on item $I_j$ completes at time $t$ **then**
        Let $\Delta t$ be the service time of this verification (exponential with rate $\mu_{ij}$)
        $S_{ij} \leftarrow S_{ij} + \Delta t$
        Remove this feedback from queue $Q_j(t)$ (FCFS within $Q_j$)
    **end if**
    **for** each verifier $V_i$ that is **idle** at time $t$ **do**
        *// Total busy time of $V_i$ since last initialization:*
        $t_i \leftarrow \sum_{j=1}^K S_{ij}$
        *// Update deficits for items in current ambiguous set $\mathcal{B}(t)$:*
        **for** each item $j \in \mathcal{B}(t)$ **do**
            $\theta_{ij} \leftarrow x_{ij}^*(\mathcal{B}(t)) \cdot t_i - S_{ij}$
        **end for**
        *// Candidate items that $V_i$ is supposed to serve (and that have waiting feedback):*
        $\mathcal{J}_i \leftarrow \{j \in \mathcal{B}(t) \ : \ x_{ij}^*(\mathcal{B}(t)) > 0 \text{ and } Q_j(t) > 0\}$
        **if** $\mathcal{J}_i \neq \emptyset$ **then**
            Select $j^* \in \mathcal{J}_i$ such that
                $\theta_{ij^*} = \max_{j \in \mathcal{J}_i} \theta_{ij}$
            Assign verifier $V_i$ to verify the **oldest** feedback in queue $Q_{j^*}$
        **else**
            $V_i$ remains idle (no eligible job in $\mathcal{B}(t)$)
        **end if**
    **end for**
**end while**