# OpenReview forum: "ONLINE RANKING WITH UNFAIR FEEDBACK AND HUMAN VERIFICATION: HIERARCHICAL ELIMINATION AND REGRET BOUNDS"
_ICLR.cc/2026/Conference — Submitted to ICLR 2026_

### Official Review · Reviewer_sLp3 · 2025-10-27

**Soundness:** 2
**Presentation:** 3
**Contribution:** 2
**Rating:** 2
**Confidence:** 2

**Summary:**

This paper studies online ranking when user feedback can be unfair and only a limited portion of the feedback can be verified by humans with delay. The authors model the process as a continuous-time online learning problem with verification queues and propose two algorithms: Hierarchical Elimination (HE) for a single verifier and Deficit Hierarchical Elimination (DHE) for multiple verifiers with heterogeneous speeds. Theoretically, the authors provided regret analysis for both algorithms. They have also conducted numerical experiments on small synthetic data to show the efficacy of their methods.

**Strengths:**

- The paper formalizes ranking with unfair feedback and verification latency as a continuous-time bandit-queue hybrid. I think the idea here is conceptually novel and has not been studied by prior works.
- The authors provided a nice description of their algorithmic ideas, which consists of both ranking and scheduling components. (However, I think it'd be better to include the main algorithm in the main body of the paper to improve clarity.) The theoretical results also appear sound and complete, with both upper and lower bound analysis, though I have not checked the proofs in detail.

**Weaknesses:**

1. I have some major concerns related to the practicality of the proposed framework/methods:
- In terms of the numerical experiments, I think they are conducted under very small synthetic setups $(K \leq 3, T \leq 2000) $ without real or large-scale data. The paper’s claims about practical relevance to real-world online platforms are therefore not empirically supported.
- The numerical comparisons are restricted to variants of the authors’ own methods and there is no comparison against other learning-to-rank methods.
- Neither HE nor DHE’s computational or queue-simulation cost is analyzed, and it is unclear whether the proposed method would be practically feasible to implement.
2. The model assumes that unfair feedback is corrected by human verifiers operating in queues with exponential service times, which in my opinion is not very realistic for modern platforms. In practice, large-scale online platforms can use automated anomaly detection or even LLM-based verification, which can handle biased or malicious feedback at far lower computational and operational cost. So I feel that the proposed model is rather limited in real-world applicability.

**Questions:**

See weaknesses.

---

> ### Author Response · Authors · 2025-11-14
>
> Thank you for your comments, and I will clarify the issues you mentioned.
> 1. We actually conducted a 50-item 10-verifier experiment in the appendix, and we understand that empirical real-world data will support the effectiveness of our algorithm, while we provide theoretical guarantees on the algorithm that we believe will also support the effectiveness.
> 2. The setting byitself is new, so it is not feasible to perform any "fair comparison" with respect to existing algorithms. But we performed additional experiments compared to simple FCFS algorithm coupled with UCB method, and there is the result showing the extra accumulated regret (percentage increase) relative to our algorithm, grouping instances by the variance of ($\mu_{ij}$):
>
> | K, T          | Low variance (<20%) | Moderate (20–50%) | High (>50%) |
> |---------------|----------------------|-------------------|-------------|
> | K=3, T=1000   | 73.59%               | 98.77%            | 235.89%     |
> | K=10, T=1000  | 130.23%              | 359.80%           | 466.12%     |
> | K=50, T=10000 | 466.19%              | 696.64%           | 1135.01%    |
> 3. The cost is very very little. The algorithm only requires solving a static LP in front once. Every other operation is simply "if else" and number calculations.
> 4. We can actually use any other service time, not necessarily exponentially distributed. While it is true that LLM can accelerate verifications, training such specific LLM is costly and there are long-lasting arguments on the correctness and bias of LLM verification. See the literature on "LLM-as-a-Judge", for example:
> [1] Shi, J., Yuan, Z., Liu, Y., Huang, Y., Zhou, P., Sun, L., & Gong, N. Z. (2024). Optimization‑based prompt injection attack to LLM‑as‑a‑judge. arXiv:2403.17710.
> [2] Shi, L., Ma, C., Liang, W., Ma, W., & Vosoughi, S. (2024). Judging the Judges: A systematic study of position bias in LLM‑as‑a‑judge. arXiv:2406.07791.
> [3] Liang, W., Yuksekgonul, M., Mao, Y., Wu, E., & Zou, J. (2023). GPT detectors are biased against non‑native English writers. Patterns, 4(7), 100779.
> ...

---

### Official Review · Reviewer_Ve2N · 2025-11-01

**Soundness:** 3
**Presentation:** 2
**Contribution:** 2
**Rating:** 6
**Confidence:** 4

**Summary:**

This paper investigates online learning to rank (OLTR) under manipulated or unreliable feedback. Motivated by real-world systems that employ human verification mechanisms, the authors model the problem as one with delayed feedback and verification-induced queuing effects. They formulate it within a continuous-time online learning framework and introduce two algorithms that effectively combine verified and unverified feedback. Theoretical analysis establishes regret guarantees and asymptotic optimality, while empirical results demonstrate the practical advantages of the proposed approach.

**Strengths:**

1) The paper tackles a timely and practically important problem arising in real-world online platforms, where ranking systems must contend with manipulated or unfair feedback and delays caused by human verification. The problem formulation captures these challenges well and is highly relevant to practical applications.
2) It introduces a novel and insightful perspective by incorporating queuing dynamics into the OLTR framework, leading to a delayed feedback model with verification-induced latency. This integration is conceptually interesting.
3) The proposed HE and DHE algorithms are theoretically sound and designed to utilize both verified and unverified feedback while accommodating verifier heterogeneity. Their design is well-motivated and grounded in theoretical analysis.

**Weaknesses:**

1) When unverified feedback is excluded, the problem reduces to a new variant of delayed feedback learning with queuing dynamics. In this case, a more thorough comparison with existing work on delayed or batched bandits would help better situate the paper’s contributions and clarify its novelty.
2) The proposed algorithms rely on a known upper bound for the probability of unfair feedback to incorporate unverified observations. This assumption may be unrealistic in practice, as such bounds are rarely available or easy to estimate. It would strengthen the work to explore adaptive approaches that can estimate or learn this parameter online.
3) The experimental evaluation is limited in scope, relying solely on synthetic simulations without validation on real-world data. In addition, the lack of comparisons with simple and intuitive baselines, such as ranking based only on verified feedback combined with standard scheduling heuristics (e.g., FCFS), makes it difficult to fully assess the empirical benefits of the proposed methods. The current setup, involving only three items and two verifiers, also appears overly simplified and may not capture the complexity of practical scenarios.

**Questions:**

1) The only experiment with a larger setting (50 items and 10 verifiers) appears in Fig. 3b of the Appendix. It would be helpful to include more experiments with larger-scale settings, especially considering recommender system applications where the number of items is typically much higher.
2) The paper does not discuss or empirically evaluate the algorithm’s time complexity. Since the proposed methods repeatedly solve LP tasks, it would be helpful to include results that assess the computational efficiency.

---

> ### Author Response · Authors · 2025-11-14
>
> Thank you for your comments, and I will answer the questions you have:
> 1. We performed additional experiments compared to a simple FCFS algorithm coupled with UCB method, and there is the result showing the extra accumulated regret (percentage increase) relative to our algorithm, grouping instances by the variance of ($\mu_{ij}$):
>
> | K, T          | Low variance (<20%) | Moderate (20–50%) | High (>50%) |
> |---------------|----------------------|-------------------|-------------|
> | K=3, T=1000   | 73.59%               | 98.77%            | 235.89%     |
> | K=10, T=1000  | 130.23%              | 359.80%           | 466.12%     |
> | K=50, T=10000 | 466.19%              | 696.64%           | 1135.01%    |
>
> 2. We actually performed numerical experiments on a larger scale, and the results are similar, and we can include them in our appendix.
> 3. The time complexity is very very little, since the LP requires only solving once in front but not each iteration. All other operations are simply computations of quantities and if-else operations.

---

### Official Review · Reviewer_kHHd · 2025-11-01

**Soundness:** 2
**Presentation:** 2
**Contribution:** 2
**Rating:** 2
**Confidence:** 3

**Summary:**

This paper investigates incorporating human verification mechanisms into online ranking algorithms to address the impact of unfair feedback. The authors tackle this problem by proposing two algorithms: Hierarchical Elimination (HE) for a single verifier and Deficit Hierarchical Elimination (DHE) for multiple heterogeneous verifiers. Both algorithms achieve logarithmic regret bounds while handling verification scheduling.

**Strengths:**

- The integration of human verification feedback into online ranking frameworks is novel and interesting.
- The algorithms consider both the use of unverified data and the presence of multiple verifiers, offering flexible solutions for different setups.

**Weaknesses:**

**Assumptions**
- Assuming that human verification can always identify the manipulation behavior is too strong. I think it is not easy to accurately judge whether a feedback is manipulated in practice.

**Theoretical analysis**
- The proof of Theorem 1 in appendix A.1 seems not complete. The authors only present the regret bound of system $H^2$, but do not show that the original system that operates under HE algorithm enjoys less regret than that of the $H^2$ system.
- The proof of Lemma 5 is unclear. The authors appear to use the Hoeffding's inequality to bound the failure probability. However, since the value $q_j(t)$ can be arbitrarily determined by an adversary, it may depend on the system’s history, making the feedback variables depend on the history feedbacks. As the Hoeffding's inequality requires the random variables to be independent, it is not applicable in this case.

**Writing**
- I'm confused about Algorithm 1. When $UCB_i < LCB_j$, the Algorithm 1 in Appendix C removes $I_i$ from both $\mathcal{A}^q$ and
$\mathcal{A}^{q+1}$. However, the description in section 4.1 says that the algorithm should send $I_i$ from $\mathcal{A}^q$ to $\mathcal{A}^{q+1}$. Which one is correct?
- In theorem 3, what does "satisfying 1" mean?
- In line 175, the paper introduces a tuple to denote the system state, but does not provide formal definition of the entries in the tuple.
- The paper should also provide the pseudocode of the DHE algorithm.

Minors:
- The titles of Appendices A.2 and A.3 should be swapped.
- It seems that the references in line 511 and line 514 are the same paper.

**Questions:**

see weakness

---

> ### Author Response · Authors · 2025-11-14
>
> Thank you for your comments in general, and I will clarify some confusing problems here:
> Assumption
> The assumption of perfect identification is indeed quite strong, but I believe it can be extended to a setting with imperfect identification by introducing an upper bound on the false identification probability. For the existing model, you can think in this way: we discard feedback that is hard to distinguish when verifying them, and in the overloaded system, it is effectively equivalent to the decrease of verification rate.
>
> Theoretical analysis
> 1. Since the system $H^1$ eliminates items earlier than uniformly sampling almost surely, which is how the system $H^2$ does. Directly analyzing $H^1$ is intractable due to state-dependent queueing (arrivals and rankings co-evolve), whereas $H^2$s simplified, slower dynamics are analyzable.
> 2. In the proof of lemma 5, we did not use any concentration inequality directly on $q_j(t)$, instead, we put a uniform upper bound to handle that part, and we only use concentration inequality on verified feedback, which is in fact independent. And it is correct that $q_j(t)$ provides zero information, so we did not use that as proof.
>
> Writings:
> 1. The description in section 4.1., i.e. "send", is correct. Thank you for pointing out the typo.
> 2. Satisfying "1" means definition 1 defined in the appendix, which is the definition of consistent policy that is commonly used in proving lower bounds. Thank you for pointing out the typo that we miss the "Definition" in front of the hyperlink "1".
> 3. We actually explain them right after it on Line 176. We only further introduce $A(t), LCB(t), UCB(t)$, which actually requires a formal definition in section 4. For the remaining ones, we believe it is clear to use just "words" to define the notation.
> 4. We will provide it.

---

> > ### Comment · Reviewer_kHHd · 2025-11-27
> >
> > Thank you for the response. However, I am still confused about the proof of Lemma 6. Could you explain in detail how Eq. (38) is derived from Eq. (37)? I also suggest that the authors highlight the revised sentences in the updated manuscript, as it is currently unclear which parts of the paper have been modified.

---

> > > ### Author Response · Authors · 2025-11-28
> > >
> > > Thank you for your response,
> > > After careful recheck of the proof, we indeed require independence of q_j(t), while it is unknown and time-variant. Therefore, it should be a simple concentration inequality for that step. I have also modified the choice of words in the corresponding paragraphs.
> > >
> > > Regarding the modification of the paper, I include the full algorithm for both HE and DHE, and also include the part of proving the system H^1 and the original system.

---

### Author Response · Authors · 2025-11-14

I have updated the revised version, and hopefully it resolves most of the concerns you raised before in the comments.

---

### Meta-Review · Area_Chair_eZNP · 2025-12-22

**Summary:**

The paper studies online learning to rank with unreliable feedback and under availability of human verification. All reviewers acknowledge that the paper idea is novel and worth pursuing.

However, reviewers have pointed our concerns about clarity (reviewers kHHd and sLp3), proof details (reviewer kHHd) and comparison with prior work (reviewers Ve2N and sLp3). They also identified some of the assumptions in the paper as too strong (all reviewers).

Overall, the paper seems to explore an interesting topic, but would benefit from further polishing and looking into relaxing at least some assumptions.

**Reviewer Concerns:**

The authors prepared a rebuttal that seems to address some of the reviewers' concerns. In particular, they responded to questions related to clarify, about the proofs (for Reviewer kHHd) and reported additional experimental results for Reviewers Ve2N and sLp3.

However, there is little to no discussion regarding the modelling assumptions and their practicality.

**Reviewer Scores:**

Only reviewer kHHd responded before discussions were interrupted. However, they had outstanding questions regarding one of the proofs.

Overall, it does not seem like the rebuttal would have led to a substantial increase in the reviewers' scores. This is in particular because of the unaddressed questions about modelling assumptions.

---

### Decision · Program_Chairs · 2026-01-26

Reject